# Uni-O4: Unifying Online and Offline Deep Reinforcement Learning with Multi-Step On-Policy Optimization

**Kun Lei**[1]    **Zhengmao He**[14*] **Chenhao Lu**[2*]  **Kaizhe Hu**[12]    **Yang Gao**[123]    **Huazhe Xu**[123]
[1] Shanghai Qi Zhi Institute.   [2] Tsinghua University, IIIS.  [3] Shanghai AI Lab.
[4] The Hong Kong University of Science and Technology (Guangzhou).
leikun980116@gmail.com, huazhe_xu@mail.tsinghua.edu.cn

## Abstract

Combining offline and online reinforcement learning (RL) is crucial for efficient and safe learning. However, previous approaches treat offline and online learning as separate procedures, resulting in redundant designs and limited performance. We ask: *Can we achieve straightforward yet effective offline and online learning without introducing extra conservatism or regularization?* In this study, we propose Uni-O4, which utilizes an on-policy objective for both offline and online learning. Owning to the alignment of objectives in two phases, the RL agent can transfer between offline and online learning seamlessly. This property enhances the flexibility of the learning paradigm, allowing for arbitrary combinations of pre-training, fine-tuning, offline, and online learning. In the offline phase, specifically, Uni-O4 leverages diverse ensemble policies to address the mismatch issues between the estimated behavior policy and the offline dataset. Through a simple offline policy evaluation (OPE) approach, Uni-O4 can achieve multi-step policy improvement safely. We demonstrate that by employing the method above, the fusion of these two paradigms can yield superior offline initialization as well as stable and rapid online fine-tuning capabilities. Through real-world robot tasks, we highlight the benefits of this paradigm for rapid deployment in challenging, previously unseen real-world environments. Additionally, through comprehensive evaluations using numerous simulated benchmarks, we substantiate that our method achieves state-of-the-art performance in both offline and offline-to-online fine-tuning learning. Our website: https://lei-kun.github.io/uni-o4/

## 1 Introduction

Imagine a scenario where a reinforcement learning robot needs to function and improve itself in the real world, the policy of the robot might go through the pipeline of training online in a simulator, then offline with real-world data, and lastly online in the real world. However, current reinforcement learning algorithms usually focus on specific stages of learning, which sophisticates the effort to train robots with a single unified framework.

Online RL algorithms require a substantial amount of interaction and exploration to attain strong performance, which is prohibitive in many real-world applications. Offline RL, in which agents learn from a fixed dataset generated by other behavior policies, is a potential solution. However, the policy trained purely by offline RL usually fails to acquire optimality due to limited exploration or poor out-of-distribution (OOD) value estimation.

A natural idea is to combine offline and online RL, which entails a common paradigm of using an offline RL algorithm to warm-start the policy and the value function with a subsequent online RL stage to further boost the performance. Although the paradigm of pre-training and fine-tuning is widely adopted in other machine learning domains such as computer vision (He et al., 2022) and natural language processing (Devlin et al., 2019), its direct application to RL is non-trivial. The reasons can be summarized as follows: firstly, offline RL algorithms require regularization

---

*Equal contribution.

(referred to as conservatism (Kumar et al., 2020) or policy constraints (Kostrikov et al., 2021)) for RL algorithms to avoid erroneous generalization to OOD data. However, when the policies and value functions trained with these regularizations are used as initialization for online learning, they could lead to fine-tuning instability due to the distribution shift. Previous works have attempted to address these challenges by employing conservative learning, introducing additional policy constraints (Nair et al., 2020; Kostrikov et al., 2021; Zheng et al., 2022; Wu et al.), $Q$-ensemble (Lee et al., 2021; Zhao et al., 2023b), or incorporating other value or policy regularization terms (Nakamoto et al., 2023; Zhang et al., 2023; Li et al., 2023) during the online learning stage. However, these methods may still suffer from an initial performance drop or asymptotical suboptimality (Wu et al.; Nakamoto et al., 2023; Li et al., 2023) in the online stage due to the conservatism inherited from offline RL training.

The objective of offline-to-online RL algorithms is to strike a trade-off between fine-tuning stability and asymptotical optimality. Challenges arise from the inherent conservatism during the offline stage and the difficulties associated with off-policy evaluation during the offline-to-online stage. To provide a clearer understanding, we track the average values of the $V$ and $Q$ functions during fine-tuning with four different methods. As depicted in Figure 1(b), SAC (off-policy) and CQL (conservatism) are used to fine-tune the

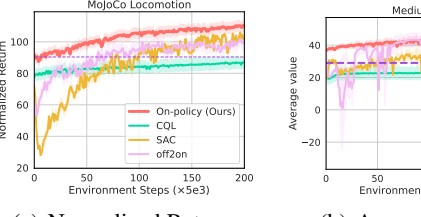

(a) Normalized Return    (b) Average value

Figure 1: (a) Normalized return curves of online fine-tuning and offline initialized scores on all Mojoco tasks. (b) Average $V$ or $Q$ -values of value functions on Hopper and Walker2d -medium tasks.

policy trained by CQL offline. Off2on (Hong et al., 2022) is a Q ensemble-based fine-tuning method. CQL shows minimal improvement over $Q$ value due to over-conservatism, leading to suboptimality. The $Q$ value of Off2on and SAC improve faster, but both of them face performance drops and unstable training during the initial stage. In contrast, Uni-O4 presents steady and rapid improvement in the $V$ values as the fine-tuning performance progresses, emerging as a favorable choice for achieving both stable and efficient fine-tuning.

Recently, the work on Behavior Proximal Policy Optimization (BPPO) (Zhuang et al., 2023) has made significant advancements by adopting the PPO (Schulman et al., 2017) objective with advantage replacement to achieve offline monotonic policy improvement. This has resulted in improved offline performance compared to one-step offline algorithms (Brandfonbrener et al., 2021). However, BPPO heavily relies on online evaluation to validate the improvement of the behavior policy, introducing risks and additional costs.

At the heart of this paper is an ***on-policy optimization method that unifies both offline and online training without extra regularization***, which we term Uni-O4. Motivated by BPPO, we employ the on-policy PPO objective to optimize both offline and online RL, aligning the objectives in both phases seamlessly. Our approach leverages an offline policy evaluation (OPE) stage to evaluate the updated policy and achieve multi-step policy improvement. OPE demonstrates appealing evaluation performance comparable to online evaluation methods. We further address the mismatch problem between the offline dataset and the estimated behavior policy by using ensemble policies and encouraging policy diversity. In this way, Uni-O4 significantly enhances the offline performance compared to BPPO, without the need for online evaluation.

After offline optimization, a standard online PPO learning stage follows to leverage the pretrained policy and value function. Such an algorithm enables a seamless transit between offline and online learning while achieving superior performance and efficiency against current approaches in both offline and offline-to-online scenarios.

To summarize, our main contribution lies in Uni-O4, which demonstrates the remarkable efficiency of the on-policy RL algorithm in offline and offline-to-online learning. Uni-O4 gets rid of the suboptimality and instability issues in current offline-to-online literature. Because of the unified design, it can be scaled up to more complex fine-tuning settings, such as online (simulator)-offline (real-world)-and online (real-world), which is useful for alleviating the sim-to-real gap in robot learning. We evaluate Uni-O4 across real-world legged robots and numerous D4RL (Fu et al., 2020) tasks. *For simulated tasks, experimental results show that Uni-O4 outperforms both SOTA offline and offline-*

*to-online RL algorithms. Furthermore, Uni-O4 excels in real-world experiments, showcasing its ability to fine-tune policies and adapt to challenging unseen environments with minimal interaction, surpassing SOTA sim2real and offline-to-online methods.*

## 2 PRELIMINARIES

**Online reinforcement learning.** We consider the Markov decision process (MDP) $\mathcal{M} = \{\mathcal{S}, \mathcal{A}, r, p, d_0, \gamma\}$, with state space $\mathcal{S}$, action space $\mathcal{A}$, reward function $r(s, a)$, transition dynamics $p$, initial state distribution $d_0(s_0)$ and discount factor $\gamma$.

In this paper, we consider on-policy RL algorithms, which typically utilize an estimator of the policy gradient and plug it into a stochastic gradient ascent algorithm. Among these algorithms, PPO (Schulman et al., 2017) introduces a conservative policy iteration term with importance sampling:

$$J_k(\pi) = \mathbb{E}_{s \sim \rho_\pi(\cdot), a \sim \pi_k(\cdot|s)} \left[ \min\left(r(\pi)A_{\pi_k}(s, a), \text{clip}\left(r(\pi), 1 - \epsilon, 1 + \epsilon\right)A_{\pi_k}(s, a)\right) \right], \quad (1)$$

where $\rho_\pi$ is the stationary distribution of states under policy $\pi$, $A_{\pi_k}(s_t, a_t)$ is the advantage function where subscript $k$ denotes the policy iteration number, $r(\pi) = \frac{\pi(a|s)}{\pi_k(a|s)}$ denotes the importance sampling ratio between the target policy $\pi$ and behavior policy $\pi_k$, $\text{clip}(\cdot)$ is a conservatism operation that constrains the ratio, and hyper-parameter $\epsilon$ is used to adjust the degree of conservatism.

**Offline reinforcement learning.** Offline RL focuses on addressing the extrapolation error due to querying the action-value function with regard to the OOD actions (Fujimoto et al., 2018; Kumar et al., 2020). Implicit Q-learning (IQL) (Kostrikov et al., 2021) revise the SARSA objective as an asymmetric $L2$ loss to estimate the maximum $Q$-value over in-distribution data, followed by an advantage-based policy extraction. Specifically, the losses of the state-value and $Q$-value functions are as follows:

$$L(V) = \mathbb{E}_{(s,a) \sim \mathcal{D}}\left[ L_2^\tau\left(\hat{Q}(s, a) - V(s)\right) \right], \quad (2)$$

$$L(Q) = \mathbb{E}_{(s,a,s') \sim \mathcal{D}}[(r(s, a) + \gamma V(s') - Q(s, a))^2], \quad (3)$$

where $\hat{Q}$ denotes the target Q function (Mnih et al., 2013), and $L_2^\tau(u)$ is the expectile loss with intensity $\tau \in [0.5, 1)$: $L_2^\tau(u) = |\tau - \mathbb{1}(u < 0)|u^2$. Based on dataset support constraints, IQL has the capability to reconstruct the optimal value function (Kostrikov et al., 2021), i.e., $\lim_{\tau \to 1} Q_\tau(s, a) = Q^*(s, a)$. In this work, we exploit the desirable property to facilitate multi-step policy optimization and recover the optimal policy. $Q_\tau$ and $V_\tau$ are the optimal solution obtained from Equation 3 and 2. We use $\widehat{Q_\tau}$ and $\widehat{V_\tau}$ to denote the value functions obtained through gradient-based optimization.

Based on one-step policy evaluation, Zhuang et al. (2023) employ an on-policy PPO objective to perform offline monotonic policy improvement:

$$J_k(\pi) = \mathbb{E}_{s \sim \rho_\mathcal{D}(\cdot), a \sim \pi_k(\cdot|s)} \left[ \min\left(r(\pi)A_{\pi_k}(s, a), \text{clip}\left(r(\pi), 1 - \epsilon, 1 + \epsilon\right)A_{\pi_k}(s, a)\right) \right]. \quad (4)$$

The only distinction between Equation 4 and 1 is that the state distribution is replaced by $\rho_\mathcal{D}(\cdot) = \sum_{t=0}^{T} \gamma^t P(s_t = s|\mathcal{D})$ and $P(s_t = s|\mathcal{D})$ is the probability of the $t$-th state equaling to $s$ in the offline dataset. Due to the advantage of this objective, BPPO is able to surpass the performance of the estimated behavior policy $\hat{\pi}_\beta$. However, they update the behavior policy $\pi_k$ by querying online evaluation, which contradicts the purpose of offline RL.

**Offline policy evaluation.** Off-policy evaluation techniques, such as fitted Q evaluation (FQE) (Paine et al., 2020), weighted importance sampling (WIS) (Voloshin et al., 2019), and approximate model (AM) (Jiang & Li, 2016), are commonly employed for model selection in offline RL. Typically, these methods require partitioning the offline dataset into training and validation sets to evaluate policies trained by offline RL algorithms. The selected hyperparameters are then used for retraining. In this work, we propose a straightforward offline policy evaluation method based on AM and one-step $Q$ evaluation without data partition and retraining. Given an offline dataset, we train an estimated dynamics model $\hat{T}$ using the maximum likelihood objective:

$$\min_{\hat{T}} \mathbb{E}_{(s,a,s) \sim \mathcal{D}}[-\log \hat{T}(s'|s, a)]. \quad (5)$$

The estimated transition model $\hat{T}$ is employed to perform $H$-step Monte-Carlo rollouts $\{s_0, a_0, \ldots, s_H, a_H\}$ for offline policy evaluation.

## 3 METHOD

In this section, we formally introduce our method, Uni-O4, which offers various training paradigms, including pure offline, offline-to-online, and online-to-offline-to-online settings. In the offline-to-online setting, our framework comprises three stages: 1) the supervised learning stage, 2) the multi-step policy improvement stage, and 3) the online fine-tuning stage, as illustrated in Figure 2.

### 3.1 ENSEMBLE BEHAVIOR CLONING WITH DISAGREEMENT-BASED REGULARIZATION

We aim to avoid extra conservatism and off-policy evaluation during both offline and offline-to-online RL learning. To this end, our entire method is built upon PPO algorithm. We start by recovering the behavior policy $\pi_\beta$ that collects the offline dataset. A straightforward approach used in Zhuang et al. (2023) is optimizing the behavior cloning objective to approximate an empirical policy $\hat{\pi}_\beta$. However, this approach leads to a mismatch in the state-action support between the estimated $\hat{\pi}_\beta$ and $\pi_\beta$ due to the presence of diverse behavior policies in the dataset $\mathcal{D}$. To address this mismatch issue, we propose an ensemble behavior cloning approach with disagreement regularization, aiming to learn diverse behaviors as initialization for policy improvement. Specifically, we learn a set of policies $\prod_n = \{\hat{\pi}_\beta^1, \ldots, \hat{\pi}_\beta^n\}$ to recover the behavior policy $\pi_\beta$. To encourage diversity among the policies, we jointly train them using the BC loss augmented by the negative disagreement penalty between each policy $\hat{\pi}_\beta^i$ and the combined policy $f(\{\hat{\pi}_\beta^j\}_{j\in[n]})$.

**Proposition 1.** *Given the dataset $\mathcal{D}$ and policies $\prod_n$, the distance over $\hat{\pi}_\beta^i(\cdot|s)$ and $f(\{\hat{\pi}_\beta^j(\cdot|s)\})$ can be expressed as $D_{KL}\left(\hat{\pi}_\beta^i(\cdot|s)||\frac{f(\{\hat{\pi}_\beta^j(\cdot|s)\})}{Z(s)}\right)$, where $Z(s)$ is the normalized coefficient.*

The average KL divergence over $\hat{\pi}_\beta^i(\cdot|s)$ and $f(\{\hat{\pi}_\beta^j(\cdot|s)\})$ can be approximated by sampling (Schulman et al., 2015a). In the offline setting, we further approximate this distance constraint by sampling actions from the dataset $\mathcal{D}$ instead of from $\hat{\pi}_\beta^i$. Meanwhile, we choose $f(\{\hat{\pi}_\beta^j(a|s)\}) = \max_{1\leqslant j\leqslant n} \hat{\pi}_\beta^j(a|s)$ motivated by Ghosh et al. (2021).

**Theorem 1.** *Given the distance $KL\left(\hat{\pi}_\beta^i(a|s)||\frac{\max_{1\leqslant j\leqslant n} \hat{\pi}_\beta^j(a|s)}{Z(s)}\right)$, we can derive the following bound: $KL\left(\hat{\pi}_\beta^i(a|s)||\frac{\max_{1\leqslant j\leqslant n} \hat{\pi}_\beta^j(a|s)}{Z(s)}\right) \geqslant \int_a \mathrm{d}a\, \hat{\pi}_\beta^i(a|s) \log \frac{\hat{\pi}_\beta^i(a|s)}{\max_{1\leqslant j\leqslant n} \hat{\pi}_\beta^j(a|s)}.$*

The proof and the definition of $Z(s)$ are presented in Appendix A.1. Then, we can enhance the diversity among behavior policies by optimizing the lower bound:

$$\text{Maximize:} J(\hat{\pi}_\beta^i) = \mathbb{E}_{(s,a)\sim\mathcal{D}}\log\hat{\pi}_\beta^i(a|s) + \alpha\mathbb{E}_{(s,a)\sim\mathcal{D}} \log\left(\frac{\hat{\pi}_\beta^i(a|s)}{\max_{1\leqslant j\leqslant n} \hat{\pi}_\beta^j(a|s)}\right), \quad (6)$$

where $\alpha$ is a hyper-parameter. In the subsequent experimental section, we empirically demonstrate that the policy ensemble can mitigate the mismatch issues between the estimated behavior policies and the offline dataset, especially for capturing the multi-modality, thereby substantially enhancing the performance of Uni-O4.

### 3.2 MULTI-STEP POLICY ENSEMBLE OPTIMIZATION

In this work, we propose a simple offline policy evaluation (OPE) method to achieve multi-step policy improvement, in contrast to the frequent online evaluations required in BPPO (Zhuang et al., 2023). In this way, Uni-O4 can better follow the setting of offline RL.

For each behavior policy, Uni-O4 performs policy gradient optimization similar to Equation 4, but we aim to achieve multi-step policy optimization via querying OPE. The clipped surrogate objective for each policy is given as follows. Since we add the iteration number as a subscript for behavior policies, we overload the notion of behavior policies as $\pi_k^i$ rather than $\hat{\pi}_\beta^i$, i.e., $\pi_0^i := \hat{\pi}_\beta^i$.

$$J_k\left(\pi^i\right) = \mathbb{E}_{s\sim\rho_\mathcal{D}(\cdot),a\sim\pi_k^i(\cdot|s)} \left[\min\left(r(\pi^i)A_{\pi_k^i}(s,a), \text{clip}\left(r(\pi^i), 1-\epsilon, 1+\epsilon\right) A_{\pi_k^i}(s,a)\right)\right], \quad (7)$$

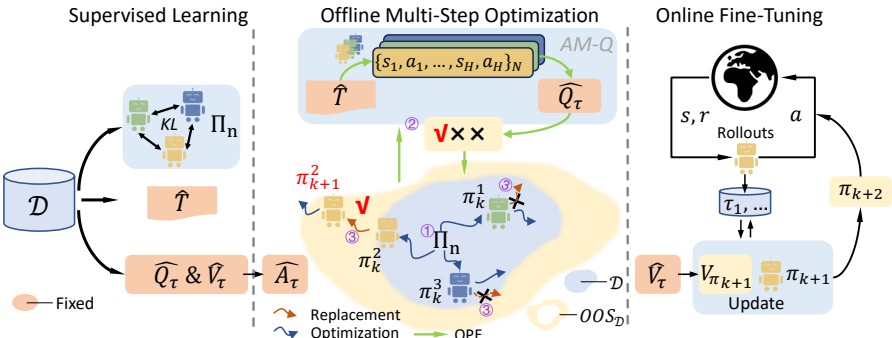

Figure 2: Uni-O4 employs supervised learning to learn the components for initializing the subsequent phase. In offline multi-step optimization phase (middle), policies query AM-Q to determine whether to replace the behavior policies after a certain number of training steps. For instance, AM-Q allows $\pi^2$ to replace its behavior policy with its target policy but rejects the others. Subsequently, one policy is selected as the initialization for online fine-tuning. Specifically, $OOS_\mathcal{D}$ indicates out-of-support of dataset.

where superscript $i$ specify the index of ensemble policies, and subscript $k$ is the iteration number, and $r(\pi^i) = \frac{\pi^i(a|s)}{\pi_k^i(a|s)}$ denotes the probability ratio between the target policy $\pi^i$ and behavior policy $\pi_k^i$. If we set $k = 0$, i.e., fixing the estimated behavior policy by Equation 6 for action sampling, this method will degenerate to one-step RL. However, this will lead to sub-optimal policies because the target policy is constrained to be close to the estimated behavior policy by the clip function and the performance is heavily dependent on the accuracy of the estimated behavior policy.

**Offline policy evaluation for multi-step policy improvement.** For achieving multi-step policy improvement safely, we update the behavior policies by querying OPE. Specifically, our OPE method combines the approximate model (**AM**) and fitted **Q** evaluation (AM-Q).

**Definition 1.** *Given the true transition model $T$ and optimal value function $Q^*$, we define the AM-Q as:* $J(\pi) = \mathbb{E}_{(s,a) \sim (T,\pi)} \left[ \sum_{t=0}^{H-1} Q^*(s_t, a_t) \right]$, *where $H$ is the horizon length.*

In the offline setting, however, the agent does not have access to the true model and $Q^*$. But $Q_\tau$ approaches $Q^*$ based on the dataset support constraints. We fit $\hat{T}$ and $\widehat{Q_\tau}$ by Equation 5 and 3. In this way, the practical AM-Q can be expressed as $\widehat{J_\tau}(\pi) = \mathbb{E}_{(s,a) \sim (\hat{T},\pi)} \left[ \sum_{t=0}^{H-1} \widehat{Q_\tau}(s_t, a_t) \right]$. Thus, the OPE bias is mainly coming from the transition model approximation.

**Theorem 2.** *Given the estimated $\hat{T}$ and $\widehat{Q_\tau}$, we can derive the following bound:* $|J(\pi, T) - J(\pi, \hat{T})| \leqslant Q_{\max} \frac{H(H-1)}{2} \sqrt{2 D_{KL}(T\pi\rho || \hat{T}\pi\rho)}$, *where we assume $\widehat{Q_\tau}$ is bounded by $Q_{\max}$.*

The proof is presented in Appendix A.2. BPPO (Zhuang et al., 2023) have derived the offline monotonical improvement bound (their Theorem 3) between the target policy and behavior policy, i.e., $\pi_{k+1}$ and $\pi_k$. However, since the behavior policy is updated iteratively, this bound can result in cumulative errors that disrupt the desired monotonicity. Consequently, this bound cannot be directly utilized in the offline setting. This limitation is the reason why BPPO requires online evaluation to ensure performance improvement when replacing the behavior policy with the target policy. Given the OPE bound, we can replace the online evaluation with AM-Q to guarantee monotonicity.

Consequently, AM-Q becomes a computationally efficient OPE method, as it only requires running all state-action pairs as input, resulting in an inference complexity of $\mathcal{O}(NH)$, where $N$ is the trajectory number and no extra fitting requirements. In this way, the behavior policy $\pi_k^i$ is replaced by the target policy $\pi^i$ and the iteration number becomes to $k+1$ if OPE satisfies $\widehat{J_\tau}(\pi^i) - \widehat{J_\tau}(\pi_k^i) > 0$. We will evaluate the accuracy of the AM-Q in the following experimental section.

We compute the advantage function by $(\widehat{Q_\tau} - \widehat{V_\tau})$ rather than the GAE (Schulman et al., 2015b) used in online PPO due to the prohibition of the interaction with environments. Intuitively, *This alternative and clip function can be naturally regarded as conservative terms for offline policy optimization*. While multi-step policy optimization performs exploration to discover the optimal policies, the advantage function and clip function constrain the policy improvement in the trust region. After the

policy improvement phase, one policy is chosen by querying OPE with minimal evaluation budgets in Appendix A.13.

**Online PPO fine-tuning.** The policy and value function obtained by offline training is directly used as initialization for a standard online PPO algorithm. The overall workflow of our algorithm is listed in Algorithm 2 of Appendix A.13. There is no extra conservative regularization or replay buffer balance strategy in the whole offline and offline-to-online training. Benefitted from the preferable properties of the on-policy algorithms, the proposed algorithm is very simple and efficient.

## 4 RELATED WORK

**Offline-to-online RL.** *PEX* (Zhang et al., 2023) introduce policy regularization through expansion and adopt a Boltzmann action selection scheme. Lee et al. (2021), Zhao et al. (2023a) and Zhao et al. (2023b) utilizes an ensemble of pessimistic value functions to address distributional shift, which can be seen as an implicit form of conservatism. Nakamoto et al. (2023) trains an additional value function to alleviate over-conservatism issues arising from the initialized value function in the offline phase. Niu et al. (2022) introduces a dynamics-aware policy evaluation scheme to solve the dynamic gap between the source and target domains. Li et al. (2023) propose a policy regularization term for trust region-style updates. Yu & Zhang (2023) leverages a standard actor-critic algorithm for aligning offline-to-online learning. Ball et al. (2023) and Zheng et al. (2023) take advantage of offline and online data for online learning. Guo et al. (2023) propose an uncertainty-guided method for efficient exploration during online fine-tuning. Also, the offline pertaining and online fine-tuning paradigm has been proved helpful in the visual RL (Ze et al., 2023a;b; Lin et al., 2023) and transformer-based offline RL methods (Zheng et al., 2022; Hu et al., 2023).

## 5 EXPERIMENTS

We conduct numerous experiments on both simulators and real-world robots to answer the following questions: **1**) Can the proposed algorithm achieve better offline RL performance and subsequently higher fine-tuning efficiency compared to previous SOTA methods? **2**) Does the algorithm lead to better asymptotic performance? **3**) Can the proposed offline policy evaluation (OPE) method provide an accurate estimation for multi-step policy improvement? **4**) Do the ensemble behavior policies provide a more comprehensive state-action support over the offline dataset for policy improvement? **5**) Can the algorithm work well on some complex tasks of real-world robot learning?

| Environment | CQL | TD3+BC | Onestep RL | IQL | COMBO | BPPO | ATAC | BC | Ours |
|---|---|---|---|---|---|---|---|---|---|
| halfcheetah-medium-v2 | 44.0 | 48.3 | 48.4 | 47.4 | **54.2** | 44.0 | **54.3** | 42.1 | 52.6±0.4 |
| hopper-medium-v2 | 58.5 | 59.3 | 59.6 | 66.3 | 97.2 | 93.9 | 102.8 | 52.8 | **104.4±0.6** |
| walker2d-medium-v2 | 72.5 | 83.7 | 81.8 | 78.3 | 81.9 | 83.6 | **91.0** | 74.0 | **90.2±1.4** |
| halfcheetah-medium-replay | 45.5 | 44.6 | 38.1 | 44.2 | **55.1** | 41.0 | 49.5 | 34.9 | 44.3±0.7 |
| hopper-medium-replay | 95.0 | 60.9 | 97.5 | 94.7 | 89.5 | 92.5 | **102.8** | 25.7 | 103.2±0.8 |
| walker2d-medium-replay | 77.2 | 81.8 | 49.5 | 73.9 | 56.0 | 77.6 | 94.1 | 18.8 | **98.4±1.6** |
| halfcheetah-medium-expert | 91.6 | 90.7 | 93.4 | 86.7 | 90.0 | 92.5 | **95.5** | 54.9 | 93.8±1.3 |
| hopper-medium-expert | 105.4 | 98.0 | 103.3 | 91.5 | 111.1 | 112.8 | **112.6** | 52.6 | 111.4±1.5 |
| walker2d-medium-expert | 108.8 | 110.1 | 113.0 | 109.6 | 103.3 | 113.1 | 116.3 | 107.7 | **118.1±2.2** |
| *locomotion total* | *698.5* | *677.4* | *684.6* | *692.4* | *738.3* | *751.0* | ***818.9*** | *463.5* | ***816.4±10.5*** |
| pen-human | 37.5 | 8.4 | 90.7 | 71.5 | 41.3* | **117.8** | 79.3 | 65.8 | 115.2±10.7 |
| hammer-human | 4.4 | 2.0 | 0.2 | 1.4 | 9.6* | 14.9 | 6.7 | 2.6 | **24.7±4.4** |
| door-human | 9.9 | 0.5 | -0.1 | 4.3 | 5.2* | 25.9 | 8.7 | 4.3 | **27.1±1.3** |
| relocate-human | 0.2 | -0.3 | 2.1 | 0.1 | 0.1* | **4.8** | 0.3 | 0.2 | 1.7±0.6 |
| pen-cloned | 39.2 | 41.5 | 60.0 | 37.3 | 24.6* | **110.8** | 73.9 | 60.7 | 101.3±19.3 |
| hammer-cloned | 2.1 | 0.8 | 2.0 | 2.1 | 3.3* | **8.9** | 2.3 | 0.4 | 7.0±0.9 |
| door-cloned | 0.4 | -0.4 | 0.4 | 1.6 | 0.27* | 6.2 | 8.2 | 0.9 | **10.2±2.6** |
| relocate-cloned | -0.1 | -0.3 | -0.1 | -0.2 | -0.2* | **1.9** | 0.8 | 0.1 | 1.4±0.2 |
| *Adroit total* | *93.6* | *52.2* | *155.2* | *118.1* | *84.2* | ***291.4*** | *180.2* | *135.0* | *288.6±40.0* |
| kitchen-complete | 43.8 | 0.0 | 2.0 | 62.5 | 3.5* | 91.5 | 2.0* | 68.3 | **93.6±2.5** |
| kitchen-partial | 49.8 | 22.5 | 35.5 | 46.3 | 1.2* | **57.0** | 0.0* | 32.5 | **58.3±3.6** |
| kitchen-mixed | 51.0 | 25.0 | 28.0 | 51.0 | 1.4* | 62.5 | 1.0* | 47.5 | **65.0±4.6** |
| *kitchen total* | *144.6* | *47.5* | *65.5* | *159.8* | *6.1* | *211.0* | *3.0* | *148.3* | ***216.9±10.7*** |
| *Total* | *936.7* | *777.1* | *905.3* | *970.3* | *828.6* | *1253.4* | *1002.1* | *746.8* | ***1322.0±61.2*** |

Table 1: Results on D4RL Gym locomotion, Adroit, and Kitchen tasks. We **bold** the best results and ours is calculated by averaging mean returns over 10 evaluation trajectories and five random seeds. Most of the results are extracted from the original papers, and * indicates that the results are reproduced by running the provided source code.

## 5.1 MAIN RESULTS

**Baselines.** We compare Uni-O4 with previous SOTA offline and offline-to-online algorithms. ***For offline RL***, we consider iterative methods like *CQL*, (Kumar et al., 2020), *TD3+BC* (Fujimoto & Gu, 2021) and ATAC (Cheng et al., 2022), onestep methods such as *Onestep RL* (Brandfonbrener et al., 2021) and *IQL* (Kostrikov et al., 2021), model-based RL approaches *COMBO* (Yu et al., 2021), and supervised learning methods (Chen et al., 2021) and (Emmons et al., 2021). ***For offline-to-online***, we include direct methods: *IQL* (Kostrikov et al., 2021), *CQL* (Kumar et al., 2020), ODT (Zheng et al., 2022), and *AWAC* (Nair et al., 2020); regularization methods: *PEX* (Zhang et al., 2023), SPOT (Wu et al.) and *Cal-ql* (Nakamoto et al., 2023); *Q*-ensemble based methods: *Off2on* (Lee et al., 2021); and *Scratch*: training PPO from scratch. ***For real-world robot tasks***, we consider using *IQL* and *walk these ways (WTW)* (Margolis & Agrawal, 2023), a strong RL method for quadrupedal locomotion, as baselines for comparison. See Apeendix A.5 for more details about the experiment settings and hyperparameter selection.

We first answer whether Uni-O4 can achieve competitive offline performance, which may serve as the initialization of online fine-tuning. We compare Uni-O4 with SOTA offline RL algorithms. As shown in Table 1, Uni-O4 outperforms all algorithms on 14 out of the 20 tasks. Notably, Uni-O4 surpasses all one-step algorithms, including one-step RL and IQL, which constrain the policy to stay close to the behavior policy. In contrast, Uni-O4 achieves multi-step policy improvement in a trust region by querying OPE, and benefits from the natural constraints, i.e., clip function and one-step policy evaluation. Additionally, Uni-O4 performs better than most iterative and model-based algorithms. While ATAC achieves comparable performance to Uni-O4 on MuJocCo locomotion tasks, it lags behind on the more challenging Adroit and Kitchen tasks. BPPO demonstrates promising results on Adroit and Kitchen tasks. However, it heavily relies on online policy evaluation for policy improvement despite being an offline method.

Table 2: Results of on Antmaze tasks. In BC column, the symbol * indicates that we use filter BC to recover the behavior policy. All results are extracted from the original paper except ours.

| Env | CQL | TD3+BC | Onestep | IQL | DT | RvS-R | RvS-G | BC | Ours |
|---|---|---|---|---|---|---|---|---|---|
| Umaze-v2 | 74.0 | 78.6 | 64. 3 | 87.5 | 65.6 | 64.4 | 65.4 | 54.6 | **93.7±3.2** |
| Umaze-diverse-v2 | **84.0** | 71.4 | 60.7 | 62.2 | 51.2 | 70.1 | 60.9 | 48.2 | 83.5±11.1 |
| Medium-play-v2 | 61.2 | 10.6 | 0.3 | 71.2 | 1.0 | 4.5 | 58.1 | 22.0* | **75.2±4.4** |
| Medium-diverse-v2 | 53.7 | 3.0 | 0.0 | **70.0** | 0.6 | 7.7 | 67.3 | 13.6* | 72.2±3.8 |
| Large-play-v2 | 15.8 | 0.2 | 0.0 | 39.6 | 0.0 | 3.5 | 32.4 | 35.3* | **64.9±2.5** |
| Large-diverse-v2 | 14.9 | 0.0 | 0.0 | 47.5 | 0.2 | 3.7 | 36.9 | 29.4* | **58.7±3.0** |
| *Total* | *303.6* | *163.8* | *61.0* | *378.0* | *118.6* | *153.9* | *321.0* | *236.7* | ***447.9±28.1*** |

To further evaluate Uni-O4, we conducted additional experiments on Antmaze tasks, which request the algorithm to deal with multi-task learning and sparse rewards. The results, presented in Table 2, demonstrate that Uni-O4 outperforms the majority of one-step, iterative, and supervised learning approaches. Overall, Uni-O4 achieves an impressive **79.4%** improvement over 26 tasks based on estimated behavior policies. These findings underscore the capability of Uni-O4 to serve as a satisfactory initialization for online fine-tuning.

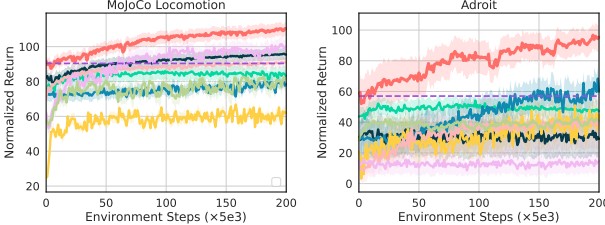

Figure 3: Aggregated learning curves of various approaches on the MuJoCo locomotion and Adroit manipulation tasks. It shares legend with Figure 4 for simplicity.

Next, we conduct experiments to demonstrate the online fine-tuning performance of Uni-O4. We run 1M environment steps for all methods. The learning curves of normalized returns are presented in Figure 3 and 4. As observed, methods that inherit conservatism and policy constraints from the offline phase, namely CQL, IQL, and AWAC, exhibit relatively stable behavior but struggle to achieve high performance, only to converge towards the offline performance of Uni-O4 on certain tasks. The policy regularization method (PEX) displays instability and initially experiences a performance drop on most tasks, failing to attain high performance compared to conservative methods. Additionally, *Q*-ensemble-based method (off2on) can be considered an implicitly conservative approach, outperforming other baselines on MuJoCo tasks but not performing well on the more challenging Adroit tasks. Moreover, these ensemble-based methods entail significant computational overhead

(*18 hours vs. 30 minutes (Uni-O4)*, see Figure 25(c) in Appendix A.12), making it unacceptable for real-world robot learning. Overall, Uni-O4 exhibits an integration of stability, consistency, and efficiency, eclipsing all baseline methods with its unified training scheme.

Figure 4: The learning curves of various methods on Adroit and MuJoCo locomotion tasks are presented across five different seeds. The solid lines indicate the mean performance, while the shaded regions represent the corresponding standard deviation.

## 5.2 APPLICATIONS ON REAL-WORLD ROBOTS

Bridging the sim-to-real gap is a widely recognized challenge in robot learning. Previous studies (Tobin et al., 2017; Yang et al., 2021; Rudin et al., 2022; Margolis & Agrawal, 2023) tackled this issue by employing domain randomization, which involves training the agent in multiple randomized environments simultaneously. However, this approach comes with computational overhead and poses challenges when applied to real-world environments that are difficult to model in simulators. To address this issue, we propose to leverage Uni-O4 in an online-offline-online framework. The agent is initially pretrained in simulators (online), followed by fine-tuning on real-world robots (offline and online), as illustrated in Figure 5(a). For more detailed information, see Appendix A.4.

In our experiments, we initiate with a policy pre-trained online in a simulator, proficient in navigating level terrains in the real world. However, when tested on a latex mattress, the policy struggled to generalize effectively to elastic terrains, resulting in overturns (first row in Figure 5(a)). The complexity arises as the material properties of the mattress pose significant challenges to simulate accurately. To counteract this, we collect offline data prior to the overturning incidents and fine-tune the policy using our method, enabling successful navigation on the mattress (second row in Figure 5(a)). Subsequently, we enhance running speed by performing online fine-tuning of this adapted policy in the real world (third row in Figure 5(a)). This methodology can be ubiquitously applied for deploying real-world solutions in environments tough to simulate, such as sandy or wetlands, showcasing the adaptability and effectiveness of our approach in handling challenging terrains, ultimately proving the superior versatility and efficacy of our method over existing solutions.

After being fine-tuned using the collected offline dataset (180,000 environment steps), the robot is able to run on the latex mattress at a low speed. Comparing the results shown in Figure 5(b), the policies trained by WTW and IQL struggle to adapt to this challenging terrain, resulting in significantly lower average returns as they are unable to move forward smoothly. However, when switching to a high-speed scenario, the policy fine-tuned by the offline dataset performs poorly, as depicted in Figure 5(c), resulting in crashes. Subsequently, after being further fine-tuned online (100,000 environment steps), the robot achieves a speed of $1.62\,\mathrm{m/s}$ (as measured by RealSense sensors). Overall, the proposed fine-tuning method can achieve significant performance improvements in both offline

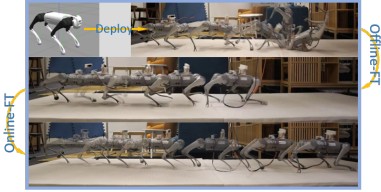

(a) Online-offline-online setting

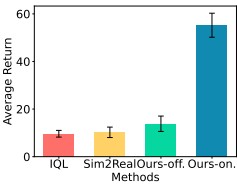

(b) Low speed testing

(c) High speed testing

Figure 5: Real-world experiments: (a) the workflow of Uni-O4 (b) Testing all methods with low-speed commands. The reported results are averaged over five trials, with each trial having a maximum of 1000 time steps. (c) Testing all methods with high-speed commands, see Appendix A.4.

and online phases, requiring only minimal interaction. In this way, offline fine-tuning ensures safety for real-world robots, while online fine-tuning continues to enhance their performance.

## 5.3 ABLATION STUDY

**OPE analysis.** To assess the accuracy of the proposed OPE method, AM-Q, we perform experiments on all MuJoCo tasks. In parallel to the execution of OPE, we conduct the online evaluation and consider it as the ground truth for calculating the accuracy of OPE estimates. Furthermore, we also tested the accuracy within a specific margin of error. As depicted in Figure 6(a), the OPE accuracy reaches approximately $80\%$. When allowing for a $20\%$ estimation error, the accuracy approaches $95\%$. This demonstrates the reliable evaluation of AM-Q for multi-step policy improvement.

**Hyper-parameter analysis.** We evaluated the disagreement penalty coefficient $\alpha$ and ensemble size in Figure 6(b) and 6(c), respectively. These two hyper-parameters were determined through experiments conducted on all MoJoCo tasks. From the results, it is evident that a minor disagreement penalty yields better results compared to having no penalty or using larger ones as behavior policy initialization. As for the ensemble size, we chose 4 as a trade-off between performance and computational efficiency, which resulted in only a small increase in training time. These findings demonstrate that the policies trained using Equation 6 offer better support over the offline dataset than a single policy. To further substantiate these phenomenons, we conduct several analyses on these two methods, as shown in Figure 12, 13, 14, 15, and 16 of Appendix A.6. The results clearly support the selection of key hyper-parameters.

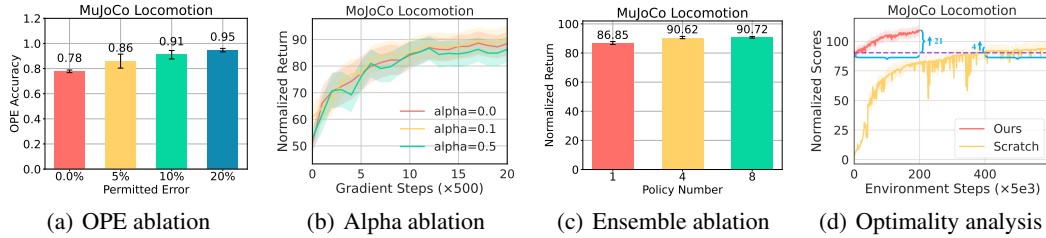

(a) OPE ablation

(b) Alpha ablation

(c) Ensemble ablation

(d) Optimality analysis

Figure 6: Ablation study on on all MoJoCo tasks, refer to Figures 26, 27, 28, and 29 in Appendix A.14 for all results. For the ablation study of key design choices of Uni-O4, see Appendix A.11.

**Optimality analysis.** As an on-policy training baseline, we running PPO for three million steps in the MoJoCo Locomotion environments. As depicted in Figure 6(d), once surpassing the offline performance, the improvement plateaus. In contrast, our fine-tuning method demonstrates rapid performance improvement, demonstrating its stable and efficient fine-tuning capabilities.

## 6 CONCLUSION

We introduce Uni-O4, which employs an on-policy algorithm to facilitate a seamless transition between offline and online learning. In the offline learning stage, a policy ensemble enjoys multi-step policy improvement by querying a proposed sample offline policy evaluation procedure. Leveraging the superior offline initialization, a standard online policy gradient algorithm continuously enhances performance monotonically. The advantageous property of the on-policy algorithm allows Uni-O4 to scale effectively to various offline and online fine-tuning scenarios across different tasks, making it a promising candidate for various offline-to-online real robot tasks.

## ACKNOWLEDGMENTS

We highly appreciate Jiacheng You for his valuable suggestions and discussions on Theorems 1 and 2, and Zifeng Zhuang for the helpful discussions during the rebuttal round.

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

## A APPENDIX

### A.1 PROOF OF THEOREM 1

$Proof.$ First let:

$$Z(s) = \int_a da \max_{1 \leqslant j \leqslant n} \pi_j(a|s), \tag{8}$$

then we have:

$$Z(s) \leqslant \int_a da \sum_{j=1}^n \pi_j(a|s) = \sum_{j=1}^n \int_a da \, \pi_j(a|s) = \sum_{j=1}^n 1 = n, \tag{9}$$

and

$$Z(s) \geqslant \int_a da \frac{1}{n} \sum_{j=1}^n \pi_j(a|s) = 1 \tag{10}$$

Based on Equation 9 and 10, we can derive the following bound:

$$
\begin{aligned}
\mathrm{KL}\left(\pi_i(a|s) || \frac{\max_{1 \leqslant j \leqslant n} \pi_j(a|s)}{Z(s)}\right) &= \int_a da \, \pi_i(a|s) \log \frac{\pi_i(a|s)}{\frac{\max_{1 \leqslant j \leqslant n} \pi_j(a|s)}{Z(s)}} \\
&= \int_a da \, \pi_i(a|s) \left(\log \frac{\pi_i(a|s)}{\max_{1 \leqslant j \leqslant n} \pi_j(a|s)} + \log Z(s)\right) \\
&\geqslant \int_a da \, \pi_i(a|s) \left(\log \frac{\pi_i(a|s)}{\max_{1 \leqslant j \leqslant n} \pi_j(a|s)} + \log 1\right) \\
&= \int_a da \, \pi_i(a|s) \log \frac{\pi_i(a|s)}{\max_{1 \leqslant j \leqslant n} \pi_j(a|s)},
\end{aligned}
\tag{11}
$$

and

$$\mathrm{KL}\left(\pi_i(a|s) || \frac{\max_{1 \leqslant j \leqslant n} \pi_j(a|s)}{Z(s)}\right) \leqslant \int_a da \, \pi_i(a|s) \left(\log \frac{\pi_i(a|s)}{\max_{1 \leqslant j \leqslant n} \pi_j(a|s)}\right) + \log n \tag{12}$$

### A.2 PROOF OF THEOREM 2

$Proof.$ We first define AM-Q as:

$$J(\pi) = \mathbb{E}_{(s,a) \sim (T,\pi)} \left[ \sum_{t=0}^{H-1} Q^*(s_t, a_t) \right] \tag{13}$$

In this work, we use the optimal solution $Q_\tau$ to approximate $Q^*$ due to $\lim_{\tau \to 1} Q_\tau(s,a) = Q^*(s,a)$. However, we just can get $\widehat{Q_\tau}$ by gradient-based optimization. And, the agent does not have access to the true model. Thus, the practical AM-Q is:

$$\widehat{J_\tau}(\pi) = \mathbb{E}_{(s,a) \sim (\hat{T},\pi)} \left[ \sum_{t=0}^{H-1} \widehat{Q_\tau}(s_t, a_t) \right], \tag{14}$$

which can be estimated by $N$ trajectories with horizon H-steps.

Without loss of generality, let $(X, \sigma(X))$ be a measurable space and $\nu : \sigma(X) \to \mathbb{R}_{\geqslant 0}$ be a non-negative measure. This choice allows us to consider any state distribution $(X = S)$ or state-action distribution $(X = S \times A)$ denoted by $\mu$, which is a probability measure on $(X, \sigma(X))$. Furthermore, $\mu$ is absolutely continuous with respect to $\nu_X$, meaning that for any $E \in \sigma(X)$, if $\nu_X(E) = 0$, then $\mu(E) = 0$.

By applying the Radon-Nikodym theorem, we conclude that there exists a unique (up to almost everywhere equivalence) Lebesgue $\nu_X$-integrable function $f_\mu : \sigma(X) \to \mathbb{R}$ such that:

$$\mu(E) = \int_E f_\mu(x)\mathrm{d}\nu_X(x), \forall E \in \sigma(X) \tag{15}$$

Furthermore, let's consider the set $\mathcal{M}_X$ consisting of all finite signed measures on $(X, \sigma(X))$ that are absolutely continuous with respect to $\nu_X$. It can be easily verified that $\mathcal{M}_X$ forms a linear space over the field of real numbers $\mathbb{R}$.

Using the Radon-Nikodym theorem once again, for any $\psi \in \mathcal{M}_X$, there exists a $\nu_X$-integrable function $g_\psi : \sigma(X) \to \mathbb{R}$ such that

$$\psi(E) = \int_E g_\psi(x)\mathrm{d}\nu_X(x), \forall E \in \sigma(X) \tag{16}$$

Furthermore, the function $g_\psi$ is unique up to almost everywhere equivalence. Since $g_\psi$ is $\nu_X$-integrable, we can establish that $|g_\psi|$ is also $\nu_X$-integrable. As a result, we can define the norm of $\psi$ as the total variation norm.

$$\|\psi\| = \sup_{\mathcal{P}} \sum_{E \in \mathcal{P}} |\psi(E)| = \int_X |g_\psi(x)|\mathrm{d}\nu_X(x) \tag{17}$$

Here, $\mathcal{P} \subset \sigma(X)$ represents an arbitrary countable partition of $X$.

Note: If for all $E \in \sigma(X)$, $\psi(E) \geqslant 0$, then it is guaranteed that $\psi(X) = \|\psi\|$.

By definition, for any probability measure $\mu$, we have $\|\mu\| = 1$.

A policy $\pi$ can be defined as a linear operator from $\mathcal{M}_S$ to $\mathcal{M}_{S \times A}$, where specifically $p(s,a) = \pi(a|s)p(s)$.

The transition operator $T$ can be defined as a linear operator from $\mathcal{M}_{S \times A}$ to $\mathcal{M}_S$, where specifically $p(s') = \sum s, aT(s'|s,a)p(s,a)$.

Since $T$ maps any probability measure to a probability measure. For any $\mu \in \mathcal{M}_{S \times A}$, as long as $\forall E \in \sigma(S \times A), \mu(E) \geqslant 0$ and $\mu(S \times A) = \|\mu\| = 1$, it follows that $\forall E \in \sigma(S), (T\mu)(E) \geqslant 0$ and $(T\mu)(S) = \|T\mu\| = 1$

This immediately implies $\|T\| \geqslant 1$. To prove $\|T\| \leqslant 1$, let's assume there exists $\mu_0 \in \mathcal{M}_{S \times A}, \mu_0 \neq 0$, such that $\|T\mu_0\| > \|\mu_0\|$, without loss of generality, assume $\|\mu_0\| = 1$

Define $|\mu_0|$ such that for any $\forall E \in \sigma(S \times A), |\mu_0|(E) = \int_E |g_{\mu_0}(x)|d\nu_{S \times A}(x) \geqslant \max(0, \mu_0(E), -\mu_0(E))$

We have $\||\mu_0|\| = |\mu_0|(S) = 1$. Hence, $T|\mu_0|(E) \geqslant 0, \forall E \in \sigma(S)$, and$(T|\mu_0|)(S) = \|T|\mu_0|\| = 1$

Assuming $|\mu_0| - \mu_0 = 0$, we have $\mu_0(E) \geqslant 0, \forall E \in \sigma(S \times A)$ and $\|\mu_0\| = 1$. However, this contradicts the assumption that $\|T\mu_0\| > \|\mu_0\| = 1$

Let's consider $\mu_+ = \frac{|\mu_0| - \mu_0}{\||\mu_0| - \mu_0\|}$, we have $\mu_+(E) \geqslant 0, \forall E \in \sigma(S \times A)$ and $\|\mu_+\| = 1$

Similarly, we can take $\mu_- = \frac{|\mu_0| + \mu_0}{\||\mu_0| + \mu_0\|}$, satisfying $\mu_-(E) \geqslant 0, \forall E \in \sigma(S \times A)$ and $\|\mu_-\| = 1$

Therefore, $(T\mu_\pm)(E) \geqslant 0, \forall E \in \sigma(S)$. Consequently, $(T(|\mu_0| \mp \mu_0))(E) \geqslant 0, \forall E \in \sigma(S)$

Hence, we have $\pm(T\mu_0)(E) \leqslant (T|\mu_0|)(E), \forall E \in \sigma(S)$

Since $g_{T\mu_0}$ is $\nu_S$-integrable, it is also measurable.

We can define $E_+ = g_{T\mu_0}^{-1}([0, +\infty)) \in \sigma(S), E_- = g_{T\mu_0}^{-1}((-\infty, 0)) \in \sigma(S)$.

We have:

$$
\begin{aligned}
\|T\mu_0\| &= \int_S |g_{T\mu_0}(x)| \mathrm{d}\nu_S(x) \\
&= \int_{E_+} g_{T\mu_0}(x)\mathrm{d}\nu_S(x) - \int_{E_+} g_{T\mu_0}(x)\mathrm{d}\nu_S(x) \\
&= (T\mu_0)(E_+) - (T\mu_0)(E_-) \\
&\leqslant (T|\mu_0|)(E_+) + (T|\mu_0|)(E_-) \\
&= (T|\mu_0|)(E_+ \cup E_-) = (T|\mu_0|)(S) = 1
\end{aligned}
\tag{18}
$$

The contradiction arises from the inequality $\|T\mu_0\| > \|\mu_0\| = 1$ leading to the conclusion that $\|T\| = 1$.

Similarly, we can derive that $\|\pi\| = 1$. Consequently, the norm of their composition $\|\pi T\| \leqslant 1$.

Now let's consider a function $R : S \times A \to \mathbb{R}$ defined on $S \times A$. We define the performance metric $J(\pi, T)$ as follows:

$$
J(\pi, T) = \mathbb{E}_{(a_0, s_1, a_1, \ldots, s_{H-1}, a_{H-1}) \sim (\pi, T), s_0 \sim \rho} \left[ \sum_{t=0}^{H-1} R(s_t, a_t) \right]
\tag{19}
$$

Using the definition of expectation, let:

$$
\lambda = \sum_{t=0}^{H-1} (\pi T)^t \pi \rho.
\tag{20}
$$

Then, we have:

$$
J(\pi, T) = \int_{S \times A} R(s, a) \mathrm{d}\lambda(s, a)
\tag{21}
$$

Now let's consider different $T_1, T_2$ and their corresponding $\lambda_1, \lambda_2$, as well as the corresponding $g_1, g_2$ associated with $\lambda$, we have:

$$
\begin{aligned}
\|(\pi T_1)^t \pi \rho - (\pi T_2)^t \pi \rho\| &= \|((\pi T_1)^{t-1} + (\pi T_1)^{t-2}(\pi T_2) + \cdots + (\pi T_2)^{t-1})(\pi T_1 - \pi T_2)\pi \rho\| \\
&\leqslant \|(\pi T_1)^{t-1} + (\pi T_1)^{t-2}(\pi T_2) + \cdots + (\pi T_2)^{t-1}\|\|\pi T_1 \pi \rho - \pi T_2 \pi \rho\| \\
&\leqslant (\|(\pi T_1)^{t-1}\| + \cdots + \|(\pi T_2)^{t-1}\|)\|\pi T_1 \pi \rho - \pi T_2 \pi \rho\| \\
&\leqslant (\|\pi T_1\|^{t-1} + \cdots + \|\pi T_2\|^{t-1})\|\pi T_1 \pi \rho - \pi T_2 \pi \rho\| \\
&= t\|\pi T_1 \pi \rho - \pi T_2 \pi \rho\| \\
&= t\|\pi(T_1 \pi \rho - T_2 \pi \rho)\| \\
&\leqslant t\|\pi\|\|T_1 \pi \rho - T_2 \pi \rho\| \\
&= t\|T_1 \pi \rho - T_2 \pi \rho\|
\end{aligned}
\tag{22}
$$

Therefore, we have:

$$
\begin{aligned}
\|\lambda_1 - \lambda_2\| &= \left\| \sum_{t=0}^{H-1} (\pi T_1)^t \pi \rho - \sum_{t=0}^{H-1} (\pi T_2)^t \pi \rho \right\| \\
&\leqslant \left\| \sum_{t=0}^{H-1} (\pi T_1)^t \pi \rho - \sum_{t=0}^{H-1} (\pi T_2)^t \pi \rho \right\| \\
&= \left\| \sum_{t=0}^{H-1} (\pi T_1)^t \pi \rho - \sum_{t=0}^{H-1} (\pi T_2)^t \pi \rho \right\| \\
&\leqslant \sum_{t=0}^{H-1} \|(\pi T_1)^t \pi \rho - (\pi T_2)^t \pi \rho\| \\
&\leqslant \sum_{t=0}^{H-1} t \|T_1 \pi \rho - T_2 \pi \rho\| \\
&= \frac{H(H-1)}{2} \|T_1 \pi \rho - T_2 \pi \rho\|
\end{aligned}
\tag{23}
$$

Suppose that: $|R(s,a)| \leqslant R_{\max}, \forall (s,a) \in S \times A$

$$
\begin{aligned}
|J(\pi, T_1) - J(\pi, T_2)| &= \left| \int_{S \times A} R(s,a) \mathrm{d}\lambda_1(s,a) - \int_{S \times A} R(s,a) \mathrm{d}\lambda_2(s,a) \right| \\
&= \left| \int_{S \times A} R(s,a) g_1(s,a) \mathrm{d}\nu(s,a) - \int_{S \times A} R(s,a) g_2(s,a) \mathrm{d}\nu(s,a) \right| \\
&= \left| \int_{S \times A} R(s,a) (g_1(s,a) - g_2(s,a)) \mathrm{d}\nu(s,a) \right| \\
&\leqslant \int_{S \times A} |R(s,a)| |g_1(s,a) - g_2(s,a)| \mathrm{d}\nu(s,a) \\
&\leqslant R_{\max} \int_{S \times A} |g_1(s,a) - g_2(s,a)| \mathrm{d}\nu(s,a) \\
&= R_{\max} \|\lambda_1 - \lambda_2\| \\
&\leqslant R_{\max} H^2 \|T_1 \pi \rho - T_2 \pi \rho\|
\end{aligned}
\tag{24}
$$

According to Pinsker's inequality, we have: $\|T_1 \pi \rho - T_2 \pi \rho\| \leqslant \sqrt{2 D_{KL}(T_1 \pi \rho \| T_2 \pi \rho)}$. Therefore, we have: $|J(\pi, T_1) - J(\pi, T_2)| \leqslant R_{\max} \frac{H(H-1)}{2} \sqrt{2 D_{KL}(T_1 \pi \rho \| T_2 \pi \rho)}$

### A.3 RELATED WORK TO REAL-WORLD ROBOT LEARNING.

Real-world robot learning presents numerous challenges, including efficiency, safety, and autonomy. One possible approach is to pretrain in a simulator and then deploy on real robots, but this approach is hindered by the sim-to-real gap. To tackle these challenges, recent work suggests combining in-simulator pretraining with real-world fine-tuning Smith et al. (2022a) using deep reinforcement learning, or pure real-world learning (Smith et al., 2022b; 2023). Additionally, Gürtler et al. (2023) introduces a benchmark dataset for manipulation tasks, where real robot data is collected using a policy trained in simulators and then used to train a new policy using SOTA offline RL algorithms.

### A.4 DETAILED INFORMATION OF REAL-WORLD ROBOTS

Here, we present detailed information about the learning setting of Uni-O4 on the real-world robot tasks, showcasing its ability to excel in real-world robot applications. Our approach introduces a novel process of online pretraining in simulators, followed by offline and online fine-tuning on real-world robots. In the offline fine-tuning phase, specifically, we deploy the pretrained policy on real-world robots to collect datasets in more challenging environments. As a result of fine-tuning the offline dataset, the policy becomes capable of ruining at a low speed in these demanding

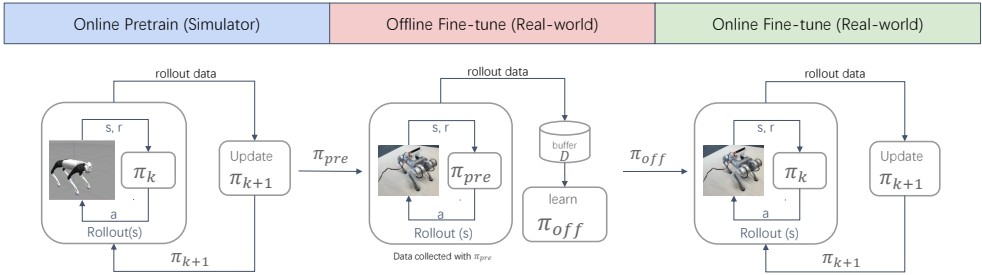

Figure 7: The workflow of our online-offline-online fine-tuning framework.

environments. Subsequently, we proceed with online fine-tuning to achieve further performance improvement. The whole workflow is presented in Figure 7. Overall, offline fine-tuning proves the safety of real-world robot learning, while online learning undergoes policy improvement. This paradigm showcases sample-efficient fine-tuning and safe robot learning.

In detail, the hardware utilized primarily includes the Unitree Go1 quadruped, RealSense T265 sensor, and a latex mattress. We deploy our algorithm on the Unitree Go1 quadruped and choose a latex mattress as the deployment environment. The RealSense T265 sensor is employed to measure the robot's speed along the x, y, and yaw axes for reward calculation. Subsequently, we provide comprehensive definitions and outline the learning process in the subsequent sections.

### A.4.1 STATE AND ACTION SPACES

In this subsection, we give the definition of the policy inputs and action space of both simulated and real-world quadruped environments in the learning process. The input to the policy is a 5-step history of observations $o$ which includes the gravity vector in the body frame $\mathbf{g}_t \in \mathbb{SO}(3)$, joint state $s_t^{\text{joint}} \in \mathbb{R}^{24}$ (joint position and velocity of each leg joint), last action $a_{t-1} \in \mathbb{R}^{12}$, the command $c_t \in \mathbb{R}^{14}$, global timing reference $t_t \in \mathbb{R}$ and the timing reference variables of 4 legs $t_{\text{leg}} \in \mathbb{R}^4$. The policy outputs the target position of the leg joint $a_t \in \mathbb{R}^{12}$, and the PD controller is then used to calculate the torque. The relative notion is listed in Table 4.

### A.4.2 REWARD FUNCTION

During pre-training, we use the same reward function as Margolis & Agrawal (2023). In the subsequent offline policy optimization and online fine-tuning stages, we use the following reward function, as shown in Table 3. We calculate the total reward as $r_{\text{pos}} \cdot \exp(c_{\text{neg}} r_{\text{neg}})$ where $r_{\text{pos}}$ is the sum of positive reward terms and $r_{\text{neg}}$ is the sum of negative reward terms (we use $c_{\text{neg}} = 0.02$). Each term is summarized in Table 3 and the relative notion is listed in Table 4.

Table 3: Reward terms for Offline and offline-to-online learning.

| Term | Expression | Weight |
|---|---|---|
| xy velocity tracking | $\exp\left\{-\left\|\mathbf{v}_{xy} - \mathbf{v}_{xy}^{\text{cmd}}\right\|^2 / \sigma_{vxy}\right\}$ | 0.02 |
| yaw velocity tracking | $\exp\left\{-\left(\boldsymbol{\omega}_z - \boldsymbol{\omega}_z^{\text{cmd}}\right)^2 / \sigma_{\omega z}\right\}$ | 0.01 |
| z velocity | $\mathbf{v}_z^2$ | $-4 \times 10^{-4}$ |
| joint torques | $\|\tau\|^2$ | $-2 \times 10^{-5}$ |
| joint velocities | $\|\dot{q}\|^2$ | $-2 \times 10^{-5}$ |
| joint accelerations | $\|\ddot{q}\|^2$ | $-5 \times 10^{-9}$ |
| action smoothing | $\|\mathbf{a}_{t-1} - \mathbf{a}_t\|^2$ | $-2 \times 10^{-3}$ |

### A.4.3 OFFLINE AND OFFLINE-TO-ONLINE LEARNING

In this section, we will provide a comprehensive explanation of the learning procedures involved in the suggested approach. As shown in Figure 7, the difference between this framework and Algorithm 2 is the behavior policies are directly initialized from the simulator pretraining rather than estimated using BC. In this manner, the policy can be trained to adapt to the challenging target environments

Table 4: Notation.

| Parameter | Definition | Units | Dimension |
|---|---|---|---|
| | Robot State | | |
| $q$ | Joint Angles | rad | 12 |
| $\dot{q}$ | Joint Velocities | rad/s | 12 |
| $\ddot{q}$ | Joint Accelerations | rad/s$^2$ | 12 |
| $\tau$ | Joint Torques | N m | 12 |
| $\mathbf{v}_{xy}$ | The velocity of the robot's xy axis. | m/s | 2 |
| $\boldsymbol{\omega}_z$ | Angular velocity of robot z axis. | rad/s | 1 |
| $\mathbf{v}_z$ | The velocity of the robot's z axis. | m/s | 1 |
| $\mathbf{v}_{xy}^{\mathrm{cmd}}$ | The command velocity of the robot's xy axis. | m/s | 2 |
| $\boldsymbol{\omega}_z^{\mathrm{cmd}}$ | Command angular velocity of robot z axis. | rad/s | 1 |
| $\mathbf{v}_z^{\mathrm{cmd}}$ | The command velocity of the robot's z axis. | m/s | 1 |
| | Control Policy | | |
| $o$ | Policy Observation | - | $58 \times 5$ |
| $a$ | Policy Action | - | 12 |

Simulator pretrained:

IQL offline trained:

Sim2Real:

Our offline fine-tuned:

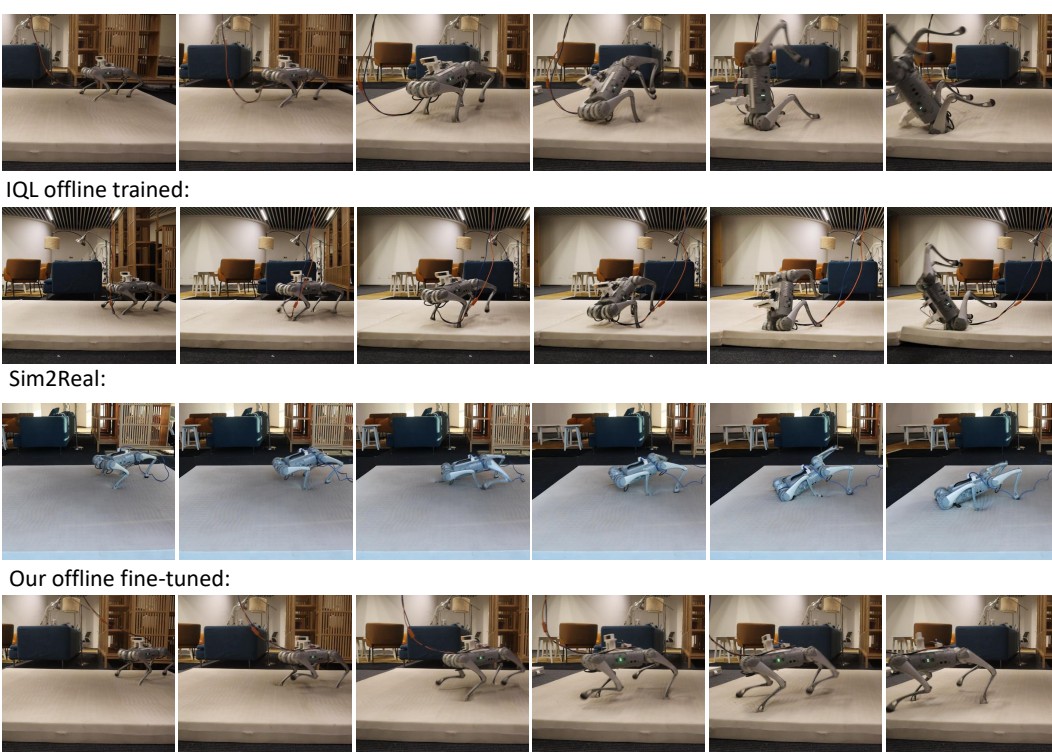

Figure 8: Deployment on a latex mattress with low speed ($1\,\mathrm{m/s}$ commands of the x aixs) by various methods. The trails from top to bottom are tested by simulator pretrained, IQL offline trained, Sim2Real, and our offline fine-tuned policies, respectively.

Sim2Real:

IQL online fine-tuned:

Ours offline fine-tuned:

Our online fine-tuned:

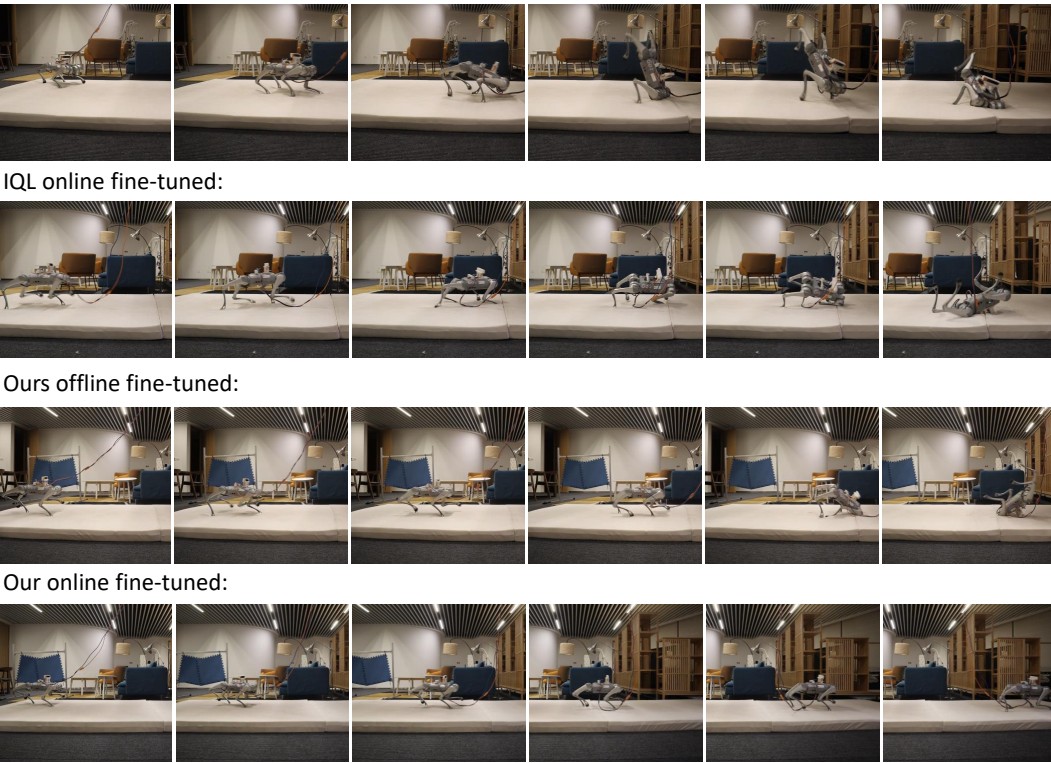

Figure 9: Deployment on a latex mattress with low speed ($2\,\mathrm{m/s}$ commands of the x aix) by various methods. The trails from top to bottom are tested by Sim2Real, IQL online fine-tuned, our offline fine-tuned, and our online fine-tuned policies, respectively.

with minimal real-world data collection. In the offline fine-tuning phase, we also encourage the policy to learn diverse behaviors. Thus, we revise Equation 7 with disagreement penalty as:

$$J_k\left(\pi^i\right) = J_k\left(\pi^i\right) + \alpha\mathbb{E}_{s\sim\mathcal{D}}[D_{\mathrm{KL}}(\hat{\pi}^i(\cdot|s)||\frac{f(\{\hat{\pi}^j(\cdot|s)\})}{Z(s)}))], \tag{25}$$

where the combined policy is the same as Equation 6.

For the online simulator pretraining, we initially trained a policy using the standard online PPO in IsaacGym, accumulating approximately seven million environment steps (training time is around 10 minutes using environment parallelism in IsaacGym). Notably, our training was solely conducted on flat ground without employing domain randomization. Subsequently, we deploy this policy on the real-world robot. As depicted in Figure 5(b) and 8, the performance on the latex mattress exhibited limitations, rendering it difficult to traverse distances greater than a few meters. Furthermore, the robot's feet were prone to sinking into the mattress, leading to instability and potential falls.

To address this challenge, we implemented offline fine-tuning by gathering approximately 180,000 step data at a frequency of 50 Hz. Each episode consisted of 1,000 steps, and we subsequently conducted offline policy optimization. The fine-tuned policy was then deployed onto the robot, enabling it to navigate the latex mattress with increased freedom at a low speed of approximately $0.8\,\mathrm{m/s}$, illustrated in Figure 8.

While the deployed policy, fine-tuned through offline methods, successfully runs in this demanding terrain, it faces limitations in tracking higher-speed commands. To overcome this, we proceed with online fine-tuning. After 100,000 step interactions, the robot achieved the ability to traverse the mattress at a higher speed ($1.6\,\mathrm{m/s}$), shown in Figure 9. In contrast, both the WTW and IQL methods exhibit limitations in adapting to the challenging environment at low or high speeds. See `uni-o4.github.io` for full videos.

### A.4.4 PSEUDO-CODE OF UNI-O4 FOR ONLINE-OFFLINE-ONLINE SETTING

In this section, we present the pseudo-code of the online-offline-to-online fine-tuning on real-world robots, outlined in Algorithm 27.

---

**Algorithm 1** On-policy policy optimization: Uni-O4

---

*# Online training stage:*
1: Initialize the policy $\pi$ and $V$-function;
2: **for** iteration $j = 1, 2, \cdots$ **do**
3:     Run policy in the target environment for $T$ timesteps
4:     Compute advantage by GAE (Schulman et al., 2015b);
5:     Update the policy by objective 1 and value by MSE loss for multiple epochs;
6: **end for**
*# Supervised learning stage:*
7: Initialize the policy ensemble $\{\pi_0^i\}_{i \in [n]}$ from online stage (set $\pi_\beta^i = \pi_0^i$);
8: Calculate value $\widehat{Q_\tau}$ and $\widehat{V_\tau}$ by 3; 2;
9: Estimate the transition model $\hat{T}$ by 5;
*# Offline policy optimization stage:*
10: **for** iteration $j = 1, 2, \cdots$ **do**
11:     Approximate advantage by $\widehat{Q_\tau} - \widehat{V_\tau}$
12:     Update each policy $\pi^i$ by maximizing objective 25;
13:     **if** $j\%C == 0$ **then**
14:         Perform OPE for each policy by AM-Q;
15:         **for** policy id $i \in 1, \ldots, n$ **do**
16:             **if** $\widehat{J_\tau}(\pi^i) > \widehat{J_\tau}(\pi_k^i)$ **then**
17:                 Set $\pi_k^i \leftarrow \pi^i$ & its $k = k + 1$;
18:             **end if**
19:         **end for**
20:     **end if**
21: **end for**
*# Online fine-tuning stage:*
22: Initialize the policy $\pi$ and $V$-function from offline stage;
23: **for** iteration $j = 1, 2, \cdots$ **do**
24:     Run policy in the target environment for $T$ timesteps
25:     Compute advantage by GAE (Schulman et al., 2015b);
26:     Update the policy by objective 1 and value by MSE loss for multiple epochs;
27: **end for**

---

### A.4.5 REAL-WORLD ROBOT BASELINE IMPLEMENTATION

We consider two baselines in the real-world robot fine-tuning setting. The first baseline is the sim-to-real work called "walk-these-way" [13]. This method focuses on quadruped locomotion and demonstrates the ability to be deployed across various terrains such as grassland, slopes, and stairs, without the need for additional training in each specific environment. This remarkable generalization capability is achieved through extensive randomization of environments and a substantial amount of training data, totaling approximately 2 billion environment steps. However, it should be noted that this method is highly data-inefficient and encounters challenges when attempting to model complex or challenging deployment environments accurately. Thus, we include this baseline in our comparison to highlight the significance of real-world fine-tuning and emphasize the sample efficiency of our online-offline-online paradigm. For WTW, we directly deploy the open-source policy trained with environment steps in the simulator.

The second baseline we consider is IQL, which is an offline-to-online method. We emphasize that IQL is regarded as a strong baseline for real-world robot fine-tining tasks (Zhou et al., 2023). Several studies (Zhou et al., 2023; Wang et al., 2023; Gürtler et al., 2023; Nair et al., 2023) have utilized IQL for fine-tuning real-world robots in an offline pretraining and online fine-tuning paradigm. In this work, we follow the offline-to-online paradigm of IQL as a baseline, aligning with these previous studies. For the IQL implementation, we begin by training IQL offline using a collected dataset for 1 million steps. We save multiple checkpoints during this training phase for evaluation purposes. Hyper-parameter tuning is performed during the offline training process. Specifically, we set $\tau = 0.7$

for value function training and $\beta = 5$ for policy extraction. Once the offline training is complete, we evaluate the checkpoints and select the one that exhibits the best performance for online fine-tuning, using the same set of hyper-parameters. To ensure fairness, we fine-tune the policy for an equal number of environment steps with Uni-O4. Following the online fine-tuning, we deploy the fine-tuned policy for comparison.

## A.5 EXPERIMENTAL SETUP DETAILS

### A.5.1 ENVIRONMENT SETTINGS

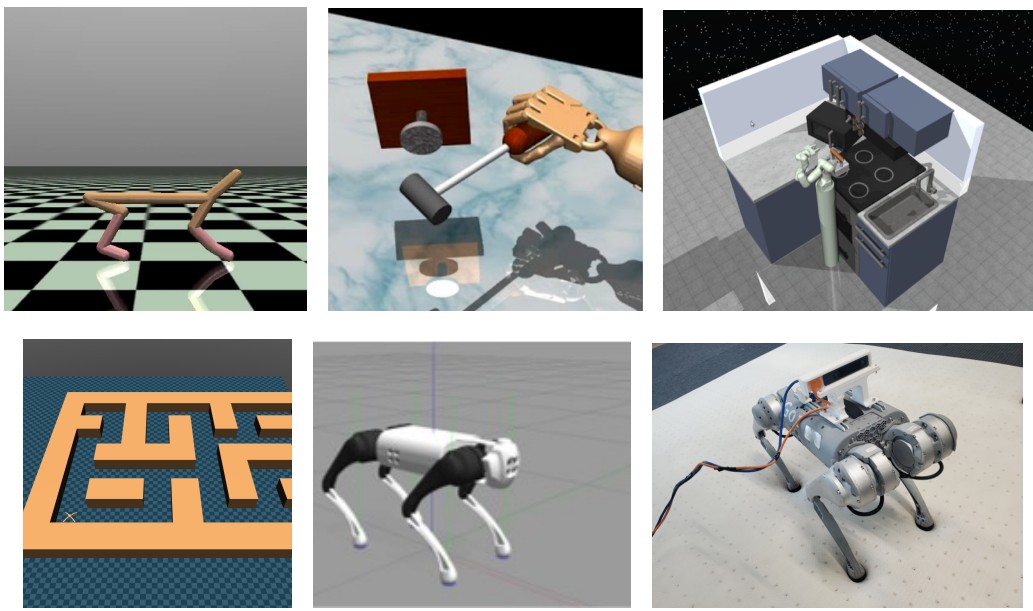

Figure 10: The suite of tasks examined in this work is illustrated successively: MoJoCo Locomotion, Adroit, Kitchen, Antmaze, simulated and real-world quadruped robots.

This study involves the evaluation of Uni-O4 using both simulated and real-world robot tasks, examining its performance in both offline and offline-to-online fine-tuning scenarios. The visualization of these domains can be seen in Figure 10. For the simulated tasks, we utilized publicly available datasets from the D4RL benchmark suite (Fu et al., 2020).

**Sim2Real quadruped robots.** We employ quadruped robots to assess the effectiveness of the proposed fine-tuning framework, which comprises three primary stages: online pretraining (simulator), offline fine-tuning (real-world), and online fine-tuning (real-world). A comprehensive description of the experimental setup and training specifics can be found in Section A.4.

**MoJoCo Locomotion Gym.** We focus on three locomotion tasks using the MuJoCo physics simulator: HalfCheetah, Walker2d, and Hopper. The objective of each task is to achieve maximum forward movement while minimizing control costs. We consider three types of datasets. The medium datasets include rollouts from medium-level policies. The medium-replay datasets comprise all samples collected during the training of a medium-level agent from scratch. Lastly, the medium-expert datasets consist of rollouts from both medium-level and expert-level policies, with an equal distribution from each.

**Adroit.** The dexterous manipulation tasks are highly challenging with sparse reward signals. The offline data used for these tasks is multi-modal, consisting of a small set of human demonstrations and a larger set of trajectories generated by a behavior-cloned policy trained on the human data. We use the rigorous evaluation criteria in Kostrikov et al. (2021) to evaluate all methods. This evaluation criteria focuses on completion speed rather than success rate. This means that the efficiency and speed at which the tasks are completed are prioritized over the mere achievement of the final goal.

**Antmaze.** In these Antmaze navigation tasks, the reward is represented by a binary variable that indicates whether the agent has successfully reached the goal or not. Once the agent reaches the goal, the episode terminates. To evaluate the performance, we use the normalized return, which is defined by Fu et al. (2020). Specifically, we conducted 50 trials to assess the agent's performance.

**Kitchen.** This environment includes various common household items such as a microwave, kettle, overhead light, cabinets, and an oven. The primary objective of each task within this domain is to interact with these items to achieve a desired goal configuration. This domain serves as a benchmark for assessing the impact of multitask behavior in a realistic non-navigation environment. **Baseline**

Table 5: Values of hyperparameters

| Hyperparameters | Values |
|---|---|
| Q network | *1024-1024* |
| V network | *256-256-256* |
| Policy network | *512-256-128* for quadruped robots
*256-256-256* for others |
| Transition model network | *200-200-200-200* for MuJoCo Locomotion tasks
*400-400-400-400* for others |
| Offline policy improvement learning rate | $1 \times 10^{-4}$ for MuJoCo Locomotion and quadruped robot tasks
$1 \times 10^{-5}$ for Adroit, Antmaze, and Kitchen tasks |
| Offline clip ratio | 0.25 |
| Online clip ratio | 0.1 |
| Online learning rate | $3 \times 10^{-5}$ for MuJoCo Locomotion and quadruped robot tasks
$8 \times 10^{-6}$ for Adroit tasks |
| Gamma | 0.99 |
| Online Lamda | 0.95 |
| Rollout steps $H$ for OPE | 1000 steps for MoJoCo Locomotion
150 steps for Antmaze
4 steps for Kitchen
20 steps for Quadruped robots |

**implementation details.** For CQL (Kumar et al., 2020), IQL (Kostrikov et al., 2021), AWAC(Nair et al., 2020), Cal-ql(Nakamoto et al., 2023), and SAC (Haarnoja et al., 2018), we use the implementation provided by Tarasov et al. (2022) with default hyperparameters. For Off2on(Lee et al., 2021), COMBO(Yu et al., 2021), ATAC(Cheng et al., 2022), and PEX(Zhang et al., 2023), we use the authors' implementation with official hyperparameters.

### A.5.2 HYPERPARAMETERS DETAILS

Here, we provide the detailed hyperparameters used in offline and offline-to-online fine-tuning phases, repetitively. We use Adam as an optimizer. For the network architectures of $Q, V, \pi$, and $\hat{T}$ are listed in Table 5.

**Offline phase.** As described in Algorithm 2, our offline learning algorithm contains two main phases: 1) the supervised learning stage; 2) the multi-step policy improvement stage. In supervised stage, we train behavior policy for $4 \times 10^5$ gradient steps using learning rate $10^{-4}$, train $Q$ and $V$ networks for $2 \times 10^6$ gradient steps using learning rate $10^{-4}$. Specifically, we update $Q$ and $V$ in the manner of IQL, thus we use the same value of coefficient $\tau$ in Kostrikov et al. (2021), i.e., $\tau = 0.9$ for Antmaze tasks and $0.7$ for others, except for *hopper-medium-expert* and *halfcheetah-medium-expert*. Because we found that $\tau = 0.5$ is best for these tasks. We train the dynamic model for $1 \times 10^6$ gradient steps with learning rate $3 \times 10^{-4}$. In the policy improvement stage, we conduct 10,000 gradient steps for each policy, in which OPE is queried per 100 steps. Learning rates are listed in Table 5.

**Online phase.** The policy and $V$ value function are initialized from offline phases. Then, we update the policy and value for $1 \times 10^6$ environment steps for the simulated tasks, and $1 \times 10^5$ environment steps for real-world robot tasks. The values of key hyperparameters are listed in Table 5. For real-world robot tasks, the hyperparameters of online simulator-based pretraining are followed by Margolis & Agrawal (2023).

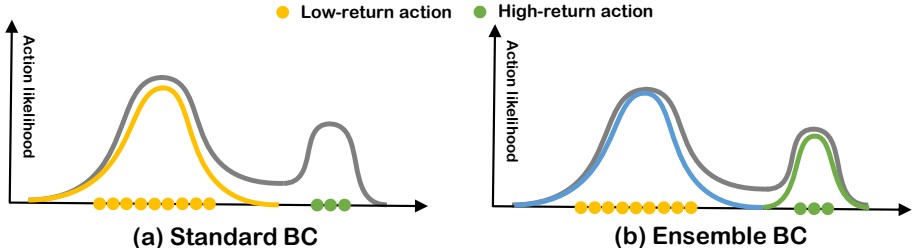

Figure 11: Motivation example of the ensemble behavior policies. The gray line represents the action distribution of the dataset ($\pi_\mathcal{D}$) which demonstrates multi-modality. The main mode is primarily composed of low-return actions, represented by the yellow dots. Conversely, the subdominant mode consists of low-density but high-return actions, denoted by the green dots. (a) Standard behavior cloning (BC) is susceptible to imitating the high-density but low-return actions, resulting in a bias towards fitting the main mode. (b) Ensemble BC approach learns diverse behavior policies that are more likely to cover all modes present in the dataset.

### A.6 Policy ensemble analysis

In Section 5.3, we conducted an analysis on the number of ensembles and the penalty hyperparameter on MoJoCo tasks. The results demonstrated that our proposed policy ensemble-based method outperforms its counterparts in terms of offline performance. To further explore the reason why diverse policies are helpful to explore higher performance policies. We provide a simple motivation example in Figure 11 to give a clearer view. There is a presence of multi-modality within a diverse dataset. In such a scenario, standard behavior cloning (BC) is susceptible to imitating the high-density but low-return actions, resulting in a bias towards fitting the main mode. However, during the offline multi-step optimization stage, the policy optimization is constrained by the clip function, making it difficult for the policy to escape this mode. Consequently, this can lead to a sub-optimal policy as it becomes unable to explore the high-return action region.

In contrast, our ensemble BC approach learns diverse behavior policies that are more likely to cover all modes present in the dataset. This facilitates exploration of the high-return region, enabling the discovery of the optimal policy. To validate our point of view, we conduct experiments to answer the following questions:

**Does ensemble policies provide better support over the offline dataset?** We visualized the relationship between state-action pairs obtained from a single policy and the offline dataset, as well as between those obtained from the combined policy ensemble used in Section 3.1 and the offline dataset. As depicted in Figure 12, we can observe that the state-action pairs supported by ensemble policies exhibit stronger correspondence with the ones projected from the offline dataset. In comparison to Figure 12(a), where the points of the two categories are more scattered, the points of two colors in Figure 12(b) have a greater overlap. This observation further confirms that the policy ensemble offers more comprehensive support over the offline dataset, ultimately leading to improved performance.

**Does behavior cloning with a disagreement penalty help in learning diverse behaviors?** We conduct an analysis to determine if the behavior cloning loss with a disagreement penalty can successfully learn diverse behavior policies. The results, depicted in Figure 13 and 14, demonstrate that the behavior policies learned using $\alpha = 0.1$ exhibit significantly greater diversity compared to their counterparts.

**Does diverse policies help in exploring optimal policies?** To answer this, we investigate whether the diverse policies help to explore the high-return actions region in the dataset, which is a crucial factor for enhancing performance in offline learning (Hong et al., 2023). We visualize the action distribution of the policies learned during the offline multi-step optimization phase on two tasks. As depicted in Figure 15 and 16, the learned policies effectively encompass the high-return actions region. This indicates that diverse policies have a greater potential for exploring optimal policies, thereby improving the overall learning process.

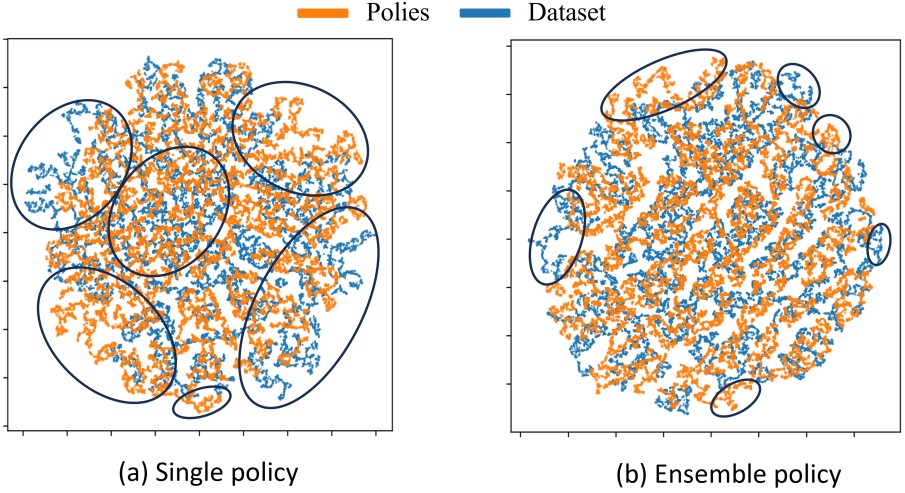

(a) Single policy          (b) Ensemble policy

Figure 12: We embed a set of state-action pairs of offline dataset and different behavior policies into a 2D space using t-SNE. We highlight the region of mismatch between the dataset and the policies.

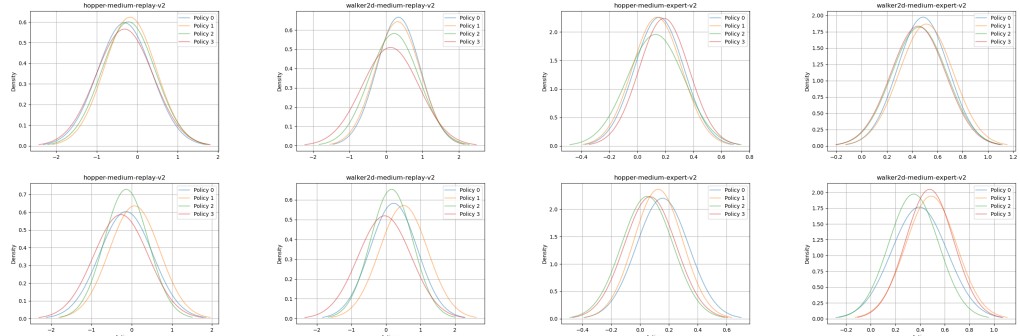

Figure 13: The distribution comparison of learned behavior policies across various disagreement penalty coefficients on the MoJoCo Locomotion domain is illustrated, specifically for $\alpha = 0.0$ (top) and $\alpha = 0.1$ (bottom). For simplicity, only the first dimension of actions is visualized.

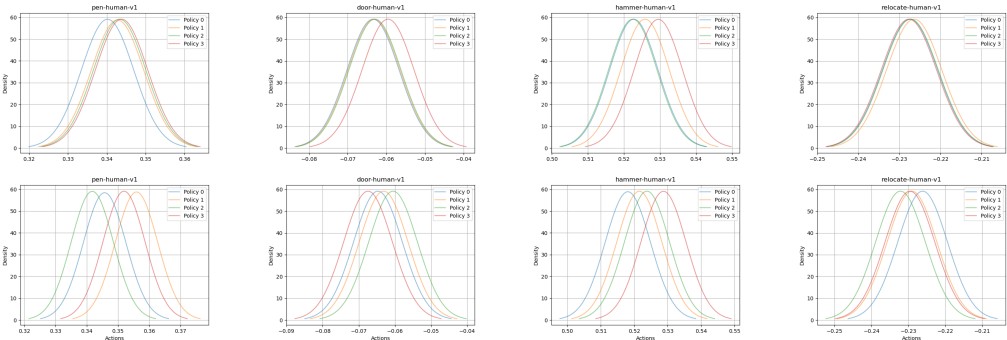

Figure 14: The distribution comparison of learned behavior policies across various disagreement penalty coefficients on the Adroit domain is illustrated, specifically for $\alpha = 0.0$ (top) and $\alpha = 0.1$ (bottom). For simplicity, only the first dimension of actions is visualized.

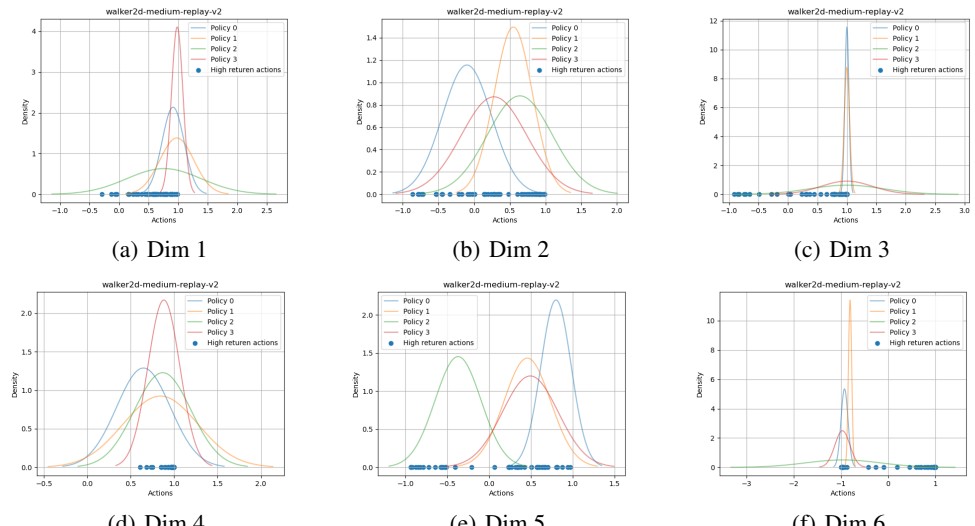

Figure 15: The distribution of each action dimension in the policies learned during the offline multi-step optimization phase on the $walker2d - medium - replay - v2$ task. Specifically, we visualize the top 50 high-return actions from the offline dataset, highlighting the diversity of policies and their ability to explore and reach the region of high-return actions.

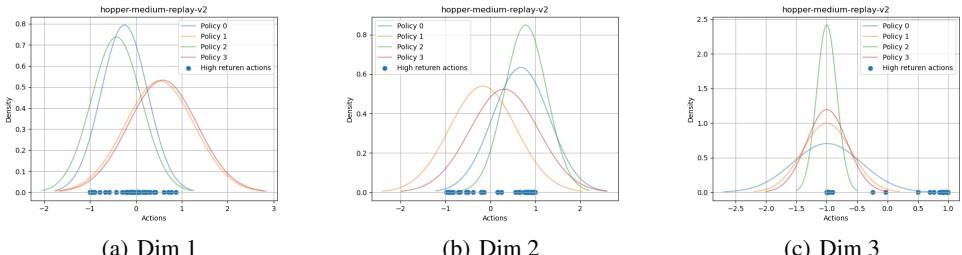

Figure 16: The distribution of each action dimension in the policies learned during the offline multi-step optimization phase on the $hopper - medium - replay - v2$ task. Specifically, we visualize the top 50 high-return actions from the offline dataset, highlighting the diversity of policies and their ability to explore and reach the region of high-return actions.

## A.7 EXTRA OFFLINE COMPARISON

In this section, we have added the Diffusion-QL (Wang et al., 2022), RORL (Yang et al., 2022), XQL (Garg et al., 2023), SQL (Xu et al., 2023) as strong baseline for comparison. Upon inspecting Table 6, it becomes evident that Uni-O4 outperforms all other methods in terms of the total score across all tasks. Furthermore, Uni-O4 outperforms all other methods in 16 out of 26 individual tasks. While RORL surpasses Uni-O4 in the total score for MoJoCo Locomotion tasks, it performs worse than Uni-O4 in the other three domains and exhibits limited effectiveness in the Kitchen domain.

Table 6: Extra comparison on D4RL tasks with other state-of-the-art baselines.

| Environment | Diffusion-QL (Wang et al., 2022) | RORL (Yang et al., 2022) | XQL (Garg et al., 2023) | SQL (Xu et al., 2023) | Ours |
|---|---|---|---|---|---|
| halfcheetah-medium-v2 | 51.1 | **66.8** | 48.3 | 48.3 | 52.6 |
| hopper-medium-v2 | 90.5 | **104.8** | 74.2 | 75.5 | **104.4** |
| walker2d-medium-v2 | 87.0 | **102.4** | 84.2 | 84.2 | 90.2 |
| halfcheetah-medium-replay | 47.8 | **61.9** | 45.2 | 44.8 | 44.3 |
| hopper-medium-replay | 101.3 | **102.8** | 100.7 | 99.7 | **103.2** |
| walker2d-medium-replay | 95.5 | 90.4 | 82.2 | 81.2 | **98.4** |
| halfcheetah-medium-expert | 96.8 | **107.8** | 94.2 | 94.0 | 93.8 |
| hopper-medium-expert | 111.1 | **112.7** | 111.2 | 111.8 | 111.4 |
| walker2d-medium-expert | 110.1 | **121.2** | 112.7 | 110.0 | 118.1 |
| *locomotion total* | 791.2 | **870.8** | 752.9 | 749.5 | *816.4* |
| Umaze-v2 | 93.4 | **96.7** | 93.8 | 92.2 | *93.7* |
| Umaze-diverse-v2 | 66.2 | **90.7** | 82.0 | 74.0 | *83.5* |
| Medium-play-v2 | 76.6 | 76.3 | 76.0 | **80.2** | *75.2* |
| Medium-diverse-v2 | **78.6** | 69.3 | 73.6 | **79.1** | *72.2* |
| Large-play-v2 | 46.4 | 16.3 | 46.5 | 53.2 | *64.9* |
| Large-diverse-v2 | 56.6 | 41.0 | 49.0 | 52.3 | *58.7* |
| *Antmaze total* | 417.8 | 390.3 | 420.9 | 431.0 | ***448.2*** |
| pen-human | 72.8 | 33.7 | 85.5 | 89.2 | **108.2** |
| hammer-human | 4.3 | 2.3 | 8.2 | 3.8 | **24.7** |
| door-human | 6.9 | 3.8 | 11.5 | 7.2 | **27.1** |
| relocate-human | 0.0 | 0.0 | 0.2 | 0.2 | **1.7** |
| pen-cloned | 57.3 | 35.7 | 53.9 | 69.8 | **101.3** |
| hammer-cloned | 2.1 | 1.7 | 4.3 | 2.1 | **7.0** |
| door-cloned | 4.1 | -0.1 | 5.9 | 4.8 | **10.2** |
| relocate-cloned | 0.0 | 0.0 | -0.2 | -0.1 | **1.4** |
| *Adroit total* | *147.5* | 77.1 | 169.3 | *177.0* | ***281.6*** |
| kitchen-complete | 84.0 | 0.3 | 82.4 | 76.4 | **93.6** |
| kitchen-partial | 60.5 | 0.0 | **73.7** | 72.5 | 58.3 |
| kitchen-mixed | 62.6 | 0.0 | 62.5 | **67.4** | 65.0 |
| *kitchen total* | *207.0* | *0.3* | **218.6** | 216.3 | *216.9* |
| *Total* | *1563.5* | *1338.5* | 1140.8 | *1573.8* | ***1763.1*** |

## A.8 EXTRA OFFLINE-TO-ONLINE RESULTS ON D4RL TASKS

In Figure 4, the results demonstrate that Uni-O4 achieves effective initialization based on suboptimal datasets for online fine-tuning. Leveraging the favorable properties of on-policy RL, Uni-O4 consistently enhances performance without any drop in performance. In this section, we conduct experiments to explore the performance of Uni-O4 on random datasets. As depicted in Figure 17, Uni-O4 exhibits rapid performance improvement with a modest initialization.

Furthermore, Figure 18 and 19 present a comparison on more challenging tasks, such as the multi-task kitchen with a long horizon and adroit hand with sparse rewards. Uni-O4 not only achieves better initialization performance but also demonstrates further performance improvements. Uni-O4 outperforms all baselines significantly, showcasing its superiority in tackling these demanding tasks.

## A.9 THE COMPARISON BETWEEN $\widehat{Q_\tau}$ AND $Q_{\pi_k}$ IN UNI-O4

In this section, we conduct experiments to investigate why we chose $\widehat{Q_\tau}$ instead of $Q_{\pi_k}$ in this work. In OPE and the computation of the advantage function, it is common to fit $Q_{\pi_k}$ during policy improvement. However, in the offline setting, both policy optimization and evaluation depend on the fitted Q-function. This can lead to overestimation due to off-policy estimation and the distribution shift present in offline RL. As depicted in Figure 20, increasing the steps of off-policy estimation

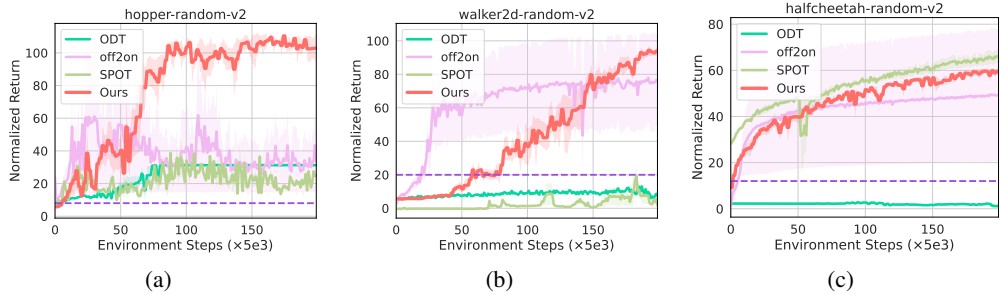

Figure 17: The experimental results on D4RL random dataset. The dotted line indicates offline initialization performance.

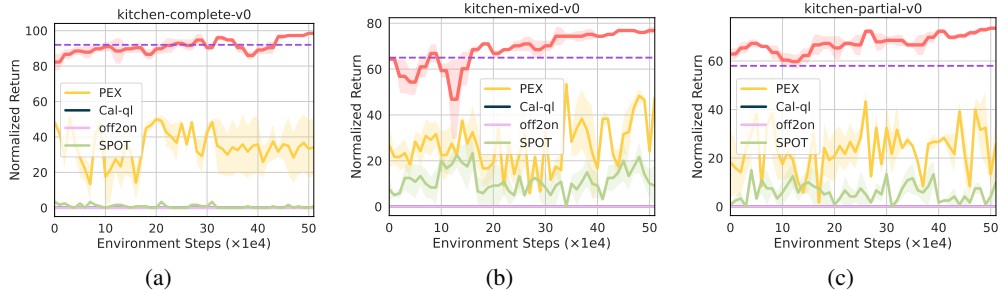

Figure 18: The experimental results on D4RL Kitchen dataset.

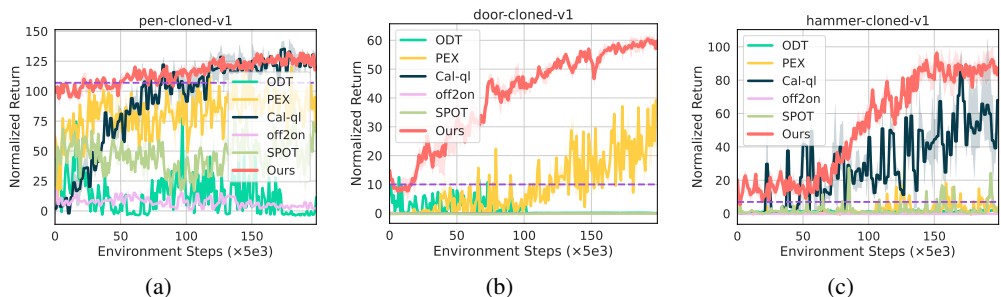

Figure 19: The experimental results on D4RL adroit cloned tasks.

results in worse performance. Conversely, using a smaller step, or even relying solely on $Q_{\pi_\beta}$, can lead to suboptimal outcomes.

On the other hand, $\widehat{Q_\tau}$ offers a favorable choice as it approximates the optimal $Q^*$ while considering the constraints imposed by the dataset support. As demonstrated, the selection of $\widehat{Q_\tau}$ significantly outperforms the iterative updating of the Q-function.

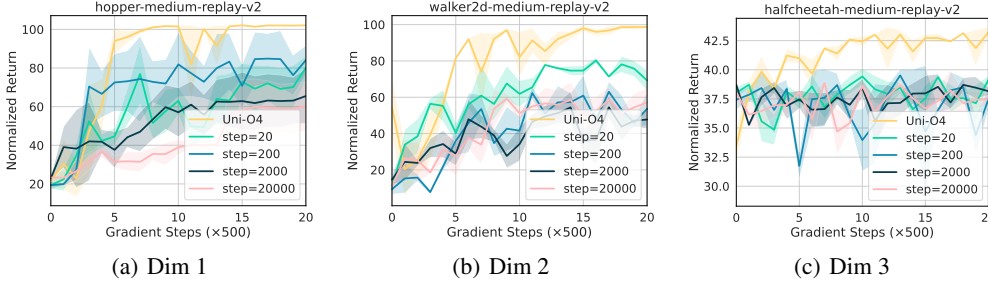

| (a) Dim 1 | (b) Dim 2 | (c) Dim 3 |

Figure 20: The comparison between $\widehat{Q_\tau}$ and $Q_{\pi_k}$ in Uni-O4. Uni-O4 represents that $\widehat{Q_\tau}$ is used as a component of AM-Q and to compute the advantage function. Additionally, the steps of 20, 200, 2000, and 20000 represent the number of iterations required to fit $\pi_k$ following the replacement of the behavior policy.

### A.10   THE DESIGN CHOICES OF ENSEMBLE BEHAVIOR POLICIES FOR MULTI-STEP POLICY OPTIMIATION

### A.11   DESIGN CHOICES ABLATION STUDY

In our evaluation, we apply the well-known 'code-level optimization' (Engstrom et al., 2019) techniques of PPO to enhance the performance of Uni-O4. Followed by the implementation of online PPO and BPPO (Zhuang et al., 2023), we use learning rate and clip ration decay, orthogonal initialization, state normalization, reward scaling, *Tanh* activation function, and mini-batch advantage normalization in Uni-O4. In this study, we specifically analyze the design choices as follows.

**Reward scaling:** Instead of directly using rewards from the environment in the objective, the PPO implementation employs a scaling scheme based on discounting. In this scheme, the rewards are divided by the standard deviation of a rolling discounted sum of the rewards, without subtracting and re-adding the mean. For more details, please refer to Engstrom et al. (2019).

**State Normalization**: Similarly to the treatment of rewards, the raw states are not directly fed into the policies. Instead, the states are normalized by the mean and variance calculated from the offline dataset instead of the initial mean-zero and variance-one vectors in standard PPO.

***Tanh* activations**: The *Tanh* activation function are used between layers in the policy.

**Value function clipping:** For value network training, we use the PPO-like objective:

$$L^V = \max\left[\left(V_{\theta_t} - V_{\text{targ}}\right)^2, \left(\text{clip}\left(V_{\theta_t}, V_{\theta_{t-1}} - \varepsilon, V_{\theta_{t-1}} + \varepsilon\right) - V_{targ}\right)^2\right],$$

where $V_\theta$ is clipped around the previous value estimates (and $\varepsilon$ is fixed to the same value as the value used to clip probability ratios in the PPO loss function.

As illustrated in Figures 21, 22, and 23, these design choices consistently exhibit enhanced performance compared to their alternatives on various tasks, including walker2d-medium, medium-replay, and medium-expert. Notably, the design choices of state normalization, reward scaling, and *Tanh* activation function show particularly significant benefits on the medium-replay tasks. Furthermore, the performance does not significantly vary with the design choice of value function clip.

### A.12   RUNNING TIME ANALYSIS

In this section, we analyze the runtime for both the offline and online phases.

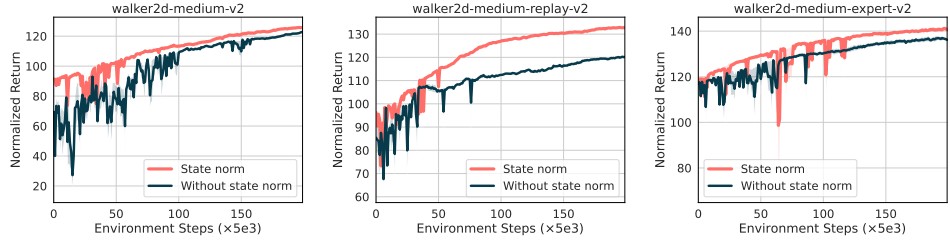

Figure 21: Ablation study on state normalization during online fine-tuning.

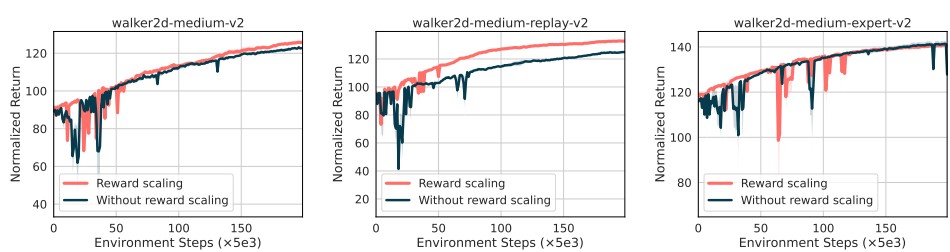

Figure 22: Ablation study on reward scaling during online fine-tuning.

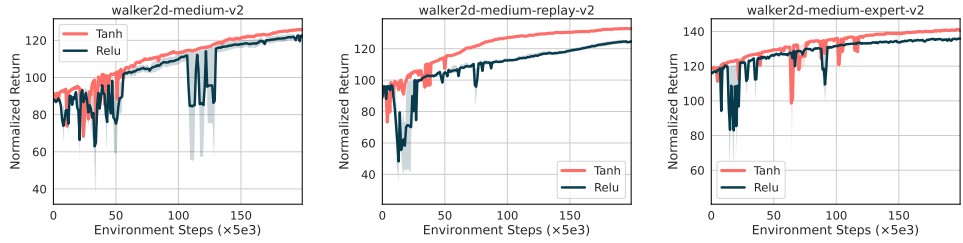

Figure 23: Ablation study on activation function during online fine-tuning.

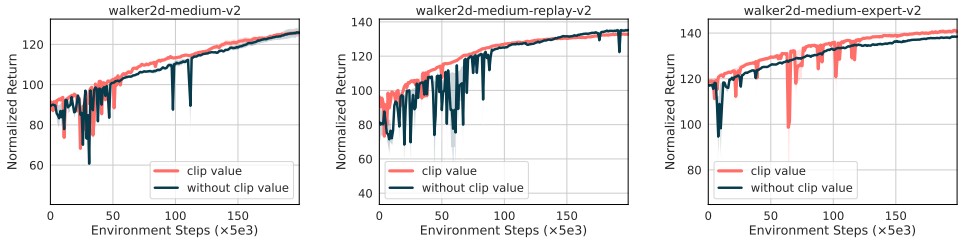

Figure 24: Ablation study on value function clip during online fine-tuning.

In the offline pretraining phase, as illustrated in Figure 25(a), the running time during offline training shows a minor increase with the number of policies. This slight increment in time is deemed acceptable, given the significant performance improvement achieved. We also measure the pretraining time for all methods in Figure 25(b). RORL, Diffusion-QL, CQL, and off2on methods require more than **7** hours, while the remaining methods complete within 5 hours.

Moving on to the online fine-tuning phase, we measure the fine-tuning time for all methods. As illustrated in Figure 25(c), the $Q$-ensemble-based method (off2on) takes over **1000** minutes, whereas Uni-O4 only requires **30** minutes to complete the fine-tuning phase. The other baselines range from approximately 200 to 400 minutes, all significantly slower than our method. These results highlight the simplicity and efficiency of Uni-O4 in terms of runtime.

All experiments are conducted on the workstation with eight NVIDIA A40 GPUs and four AMD EPYC 7542 32-Core CPUs.

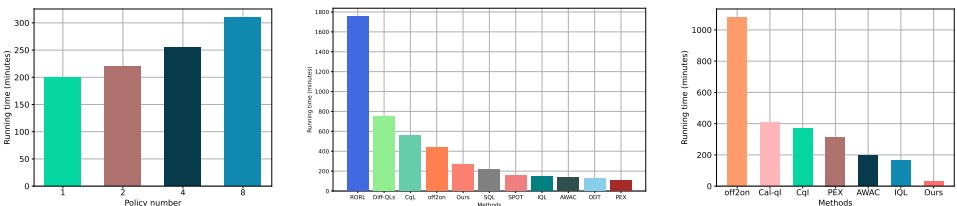

Figure 25: Running time analysis. Left: the comparison of running time over ensemble size. Middle: running time of various methods during offline pretraining. Right running time of various methods during online fine-tuning.

### A.13   PSEUDO-CODE OF UNI-O4

In this section, we present the pseudo-code for the offline-to-online stages, outlined in Algorithm 2.

To perform online fine-tuning, we select a single policy from the ensemble of policies by querying AM-Q based on its proven accurate evaluation performance, as demonstrated in Section 5.3. For instance, when allowing a 25% permitted error, the evaluation accuracy reaches approximately 95%. Alternatively, we can choose the top-$k$ policies with the highest evaluation scores. These selected policies are then evaluated through interaction with the respective environments. Specifically, we choose $k = 2$ for Mujoco locomotion and $k = 3$ for the Adroit and Kitchen domains. Based on the OPE scores, the interactions with the environments can be conducted more safely, while keeping the budgets minimal.

---

**Algorithm 2** On-policy policy optimization: Uni-O4

---

*# Supervised learning stage:*
 1: Estimate policy ensemble $\{\pi_0^i\}_{i\in[n]}$ by 6 (set $\pi_\beta^i = \pi_0^i$);
 2: Calculate value $\widehat{Q}_\tau$ and $\widehat{V}_\tau$ by 3; 2;
 3: Estimate the transition model $\hat{T}$ by 5;
*# Offline policy optimization stage:*
 4: **for** iteration $j = 1, 2, \cdots$ **do**
 5:     Approximate advantage by $\widehat{Q}_\tau - \widehat{V}_\tau$
 6:     Update each policy $\pi^i$ by maximizing objective 7;
 7:     **if** $j\%C == 0$ **then**
 8:       Perform OPE for each policy by AM-Q;
 9:       **for** policy id $i \in 1, \ldots, n$ **do**
10:         **if** $\widehat{J_\tau}(\pi^i) > \widehat{J_\tau}(\pi_k^i)$ **then**
11:           Set $\pi_k^i \leftarrow \pi^i \&$ its $k = k + 1$;
12:         **end if**
13:       **end for**
14:     **end if**
15: **end for**
*# Online fine-tuning stage:*
16: Initialize the policy $\pi$ and $V$-function from offline stage;
17: **for** iteration $j = 1, 2, \cdots$ **do**
18:     Run policy in the target environment for $T$ timesteps
19:     Compute advantage by GAE (Schulman et al., 2015b);
20:     Update the policy by objective 1 and value by MSE loss for multiple epochs;
21: **end for**

---

### A.14    FULL RESULTS OF ABLATION STUDY

Here, we provide the full results of Section 5.3. Figure 26, 27, 28, and 29 present the learning curves of ablation on OPE accuracy, hyperparameter $\alpha$, the number of ensemble policies, and the optimality on MuJoCo tasks.

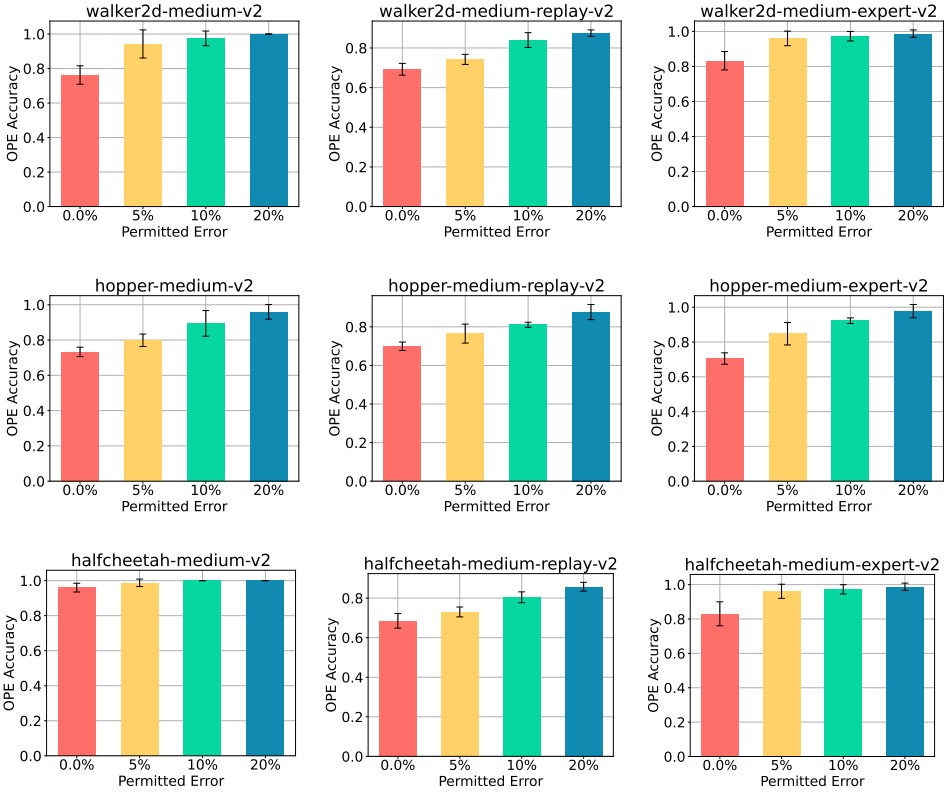

Figure 26: Full results of offline policy evaluation accuracy ablation study.

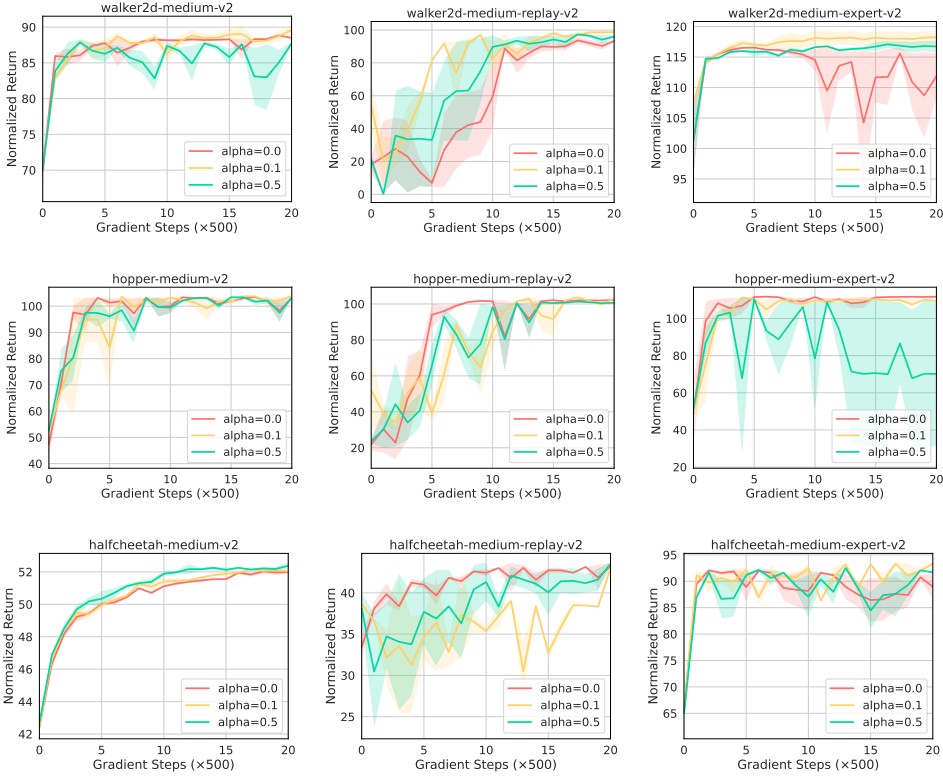

Figure 27: Full results of alpha ablation study.

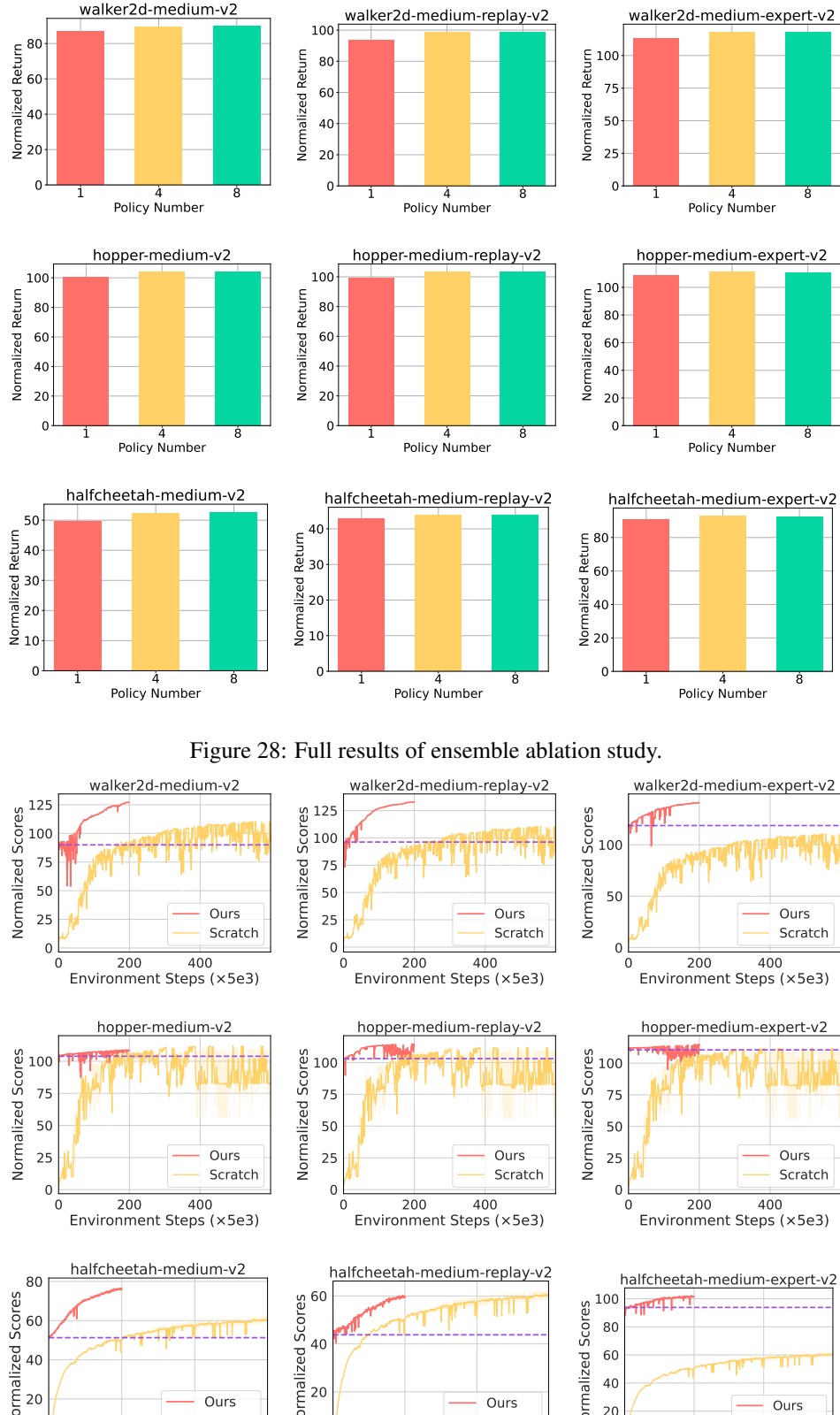

Figure 28: Full results of ensemble ablation study.

Figure 29: Full results of optimality analysis.

