# OpenReview forum: "Uni-O4: Unifying Online and Offline Deep Reinforcement Learning with Multi-Step On-Policy Optimization"
_ICLR.cc/2024/Conference — ICLR 2024 poster_

### Official Review · Reviewer_jjSi · 2023-10-20

**Soundness:** 3 good
**Presentation:** 3 good
**Contribution:** 2 fair
**Rating:** 6
**Confidence:** 5

**Summary:**

This paper focuses on unifying offline and online RL to achieve efficient and safe learning. Specifically, this paper proposes Uni-O4, which utilizes an on-policy RL objective for both offline and online learning. For offline learning, this paper combines the advantages of both BPPO and OPE to achieve the desired performance. For online learning, this paper directly utilizes the standard PPO for finetuning. Experiments under offline RL and offline-to-online RL setting demonstrate the effectiveness of Uni-O4. Furthermore, this paper extends the offline-to-online setting to address a practical robotic scenario, transforming it into an online-to-offline-to-online setting. Empirical results highlight the seamless integration across these three stages in Uni-O4.

**Strengths:**

- This paper investigates an interesting research problem: offline-to-online setting, and online(simulator)-to-offline(real-world)-to-online(real-world) setting in robotic scenarios.
- This paper performs extensive experiments to derive empirical findings.

**Weaknesses:**

Overall, this is a descent paper. However, in the current manuscript, I think the following concerns should be addressed.

== Major concern ==

- Unclear empirical motivation in Figure 1. What does these variants (Conservatism, Constraint, Off-policy) mean in (a)? How does Q value compare with V value in (b)? Moreover, from (b), it seems that CQL->SAC shows faster improvement that On-policy (V). How this conclude that Q values of SAC exhibit slow improvement? Furthermore, CQL->CQL and CQL->SAC are naïve solutions for offline-to-online RL. What about advanced offline-to-online RL algorithms, such as off2on?
- The technique seems incremental by just combining BPPO with OPE.
- I think there exhibits slight overclaiming of the experimental results in Introduction without sufficient comparison of SOTA algorithms.
> Experimental results show that Uni-O4 outperforms both SOTA offline and offline-to-online RL algorithms.
    - Insufficient comparison of offline RL, including but not limited to:

    [1] RORL: Robust Offline Reinforcement Learning via Conservative Smoothing.

    [2] Extreme Q-Learning: MaxEnt RL Without Entropy.

    [3] Offline RL with No OOD Actions: In-Sample Learning via Implicit Value Regularization.

    - Insufficient comparison (including PROTO, ODT, E2O, SPOT, etc.) or at least discussion of related works on offline-to-online RL. Particularly, the baselines include AWAC, CQL, IQL, which are naive solutions for offline-to-online RL. PEX presents weak sample-efficiency for above-medium datasets. Cal-ql is not empirically designed for MuJoCo tasks. There is only one relatively strong baseline, i.e., off2on.

    [1] Adaptive policy learning for offline-to-online reinforcement learning

    [2] Actor-Critic Alignment for Offline-to-Online Reinforcement Learning

    [3] A Simple Unified Uncertainty-Guided Framework for Offline-to-Online Reinforcement Learning

    [4] Efficient online reinforcement learning with offline data

- Minor improvement on MuJoCo tasks in Figure 4. As shown in the figure, off2on significantly outperforms Uni-O4 by a large margin in halfcheetah-medium and halfcheetah-medium-replay. Besides, I also want to point out that 100 D4RL score already achieves expert-level performance in D4RL benchmark. Thus, further improvement on other settings over 100 is not necessary. Thus, I also wonder why this work does not consider random dataset, which presents a significant challenge for online finetuning to achieve expert performance.

- Comparison in Section 5.2 seems not fair enough. Firstly, I want to know which is the claimed baseline WTW in Figure 5? Additionally, given that IQL is not designed specifically for a real-world robotic scenarios, is the comparison between IQL and Uni-O4 fair? (Uni-O4 is revised to adapt to robotic scenarios as stated in the appendix) Maybe a strong baseline can be considered to verify the superiority of Uni-O4.

- I feel a little struggling to follow Section 5.2. Maybe a pseudo-code like A.6 can be provided to make the readers understand the online-offline-online setting more clearly.

- The experimental results in A.3 make me confusing. I cannot identify obvious differences between Figure 11 (a) and (b).

== Minor Concerns ==

- Figure 2 is not that intuitive. Maybe more explanations can make it clearer.

- How many seeds and evaluation trajectories for AntMaze tasks in offline RL setting? Why offline-to-online RL setting does not consider Kitchen, AntMaze and Adroit-cloned and -relocate tasks?

- Why 18 hours training time is **unacceptable** for real-world robot learning?

- Lack of reproducibility statement.

- Maybe more details on baseline implementation for real-world robot tasks can be provided.

- Why this paper does not provide offline training time for comparison?

== Typos ==

- Page 4, above Equation 7: dataset $D$ -> $\mathcal{D}$

- Page 9, Hyper-parameter analysis, loss 7 -> Equation 7 is an optimization objective.

**Questions:**

See Weakness

---

> ### Author Response · Authors · 2023-11-20
> **Response to Reviewer Reviewer jjSi (1)**
>
> Thanks Reviewer jjSi for the valuable review of our paper. We appreciate the questions you raised and are committed to delivering a comprehensive response to address the issues.
>
> **Q1.1: What do these variants (Conservatism, Constraint, Off-policy) mean in (a)?**
>
> **A1.1:** Here we use offline -> online method to specify the offline-to-online mehtod. In the previous version of Fig. 1(a), the conservatism, constraint, and off-policy represent the offline-to-online methods CQL->CQL, IQL->IQL, and CQL->SAC. We have revised the legend to make it more clear in the updated manuscript.
>
> **Q1.2: How does the Q value compare with the V value in (b)? Moreover, from (b), it seems that CQL->SAC shows faster improvement than On-policy (V). How does this conclude that Q values of SAC exhibit slow improvement?**
>
> **A1.2:** In the motivating example, we aim to compare the naïve off-policy (SAC), conservative method (CQL), and naïve on-policy (ours). During the offline-to-online transition, naïve off-policy methods often encounter distribution shifts when the policy explores out-of-distribution regions. This can result in evaluation errors and a subsequent drop in performance. A simple method to address this challenge is to inherit the conservatism employed during the offline phase. However, relying solely on this conservative strategy can introduce suboptimality during the online phase, resulting in data inefficiency. Therefore, we aim to investigate whether a naïve on-policy RL algorithm can effectively tackle this challenge. Uni-O4 initialized the policy and $V$-function from the offline phase for a standard online PPO.
>
> *Both the $V$ and $Q$ functions can evaluate the performance of the policy. Therefore, we consider it acceptable to compare the $V$-value and $Q$-value across these methods to investigate how the value function guides performance improvement.* In the case of the naïve off-policy method, the $Q$-value exhibits a significant drop and unstable improvement during online training, leading to a decline in policy performance. The conservative method, on the other hand, shows slow improvement in the $ Q$ value, resulting in suboptimality. In contrast, the on-policy method demonstrates stable improvement in the $ V$ value, leading to consistent and efficient fine-tuning.
>
> **Q1.3: How does this conclude that Q values of SAC exhibit slow improvement?**
>
> **A1.3:** The $Q$-value of the off-policy method improves from 27 to 41, while the $V$-value of the on-policy method improves from approximately 36.5 to 47. It is true that the improvement in the $Q$-value is slightly faster than its counterpart. However, two points should be considered: *1) The on-policy method (ours) has a significantly higher initial performance from the offline phase compared to the off-policy method, resulting in higher initial scores for the value function. Therefore, a slightly slower convergence speed is reasonable. 2) The off-policy method experiences a noticeable drop and unstable improvement in the value, whereas this phenomenon is absent in the on-policy method.*
>
> **Q1.4: Furthermore, CQL->CQL and CQL->SAC are naïve solutions for offline-to-online RL. What about advanced offline-to-online RL algorithms, such as off2on?**
>
> **A1.4:** Thank you for your insightful suggestion. It is interesting to investigate how the ensemble value function guides policy improvement. Thus, we have added the off2on as a comparison. Our findings reveal that the $Q$ value of the off2on method demonstrates faster improvement and can eventually converge to the same level as the $V$ value of Uni-O4. However, it is worth noting that the off2on approach experiences performance drops and unstable training during the initial stages, primarily due to the unstable improvement of the $Q$ value.

---

> > ### Author Response · Authors · 2023-11-20
> > **Reviewer Reviewer jjSi (2)**
> >
> > **Q2: The technique seems incremental by just combining BPPO with OPE.**
> >
> > **A2**: Uni-O4 offers a significant insight by demonstrating how an on-policy RL algorithm can seamlessly unify online and offline RL, enabling flexible combinations of pretraining, fine-tuning, offline, and online learning. In the offline phase, a crucial step in offline RL, we replace online evaluation with Off-Policy Evaluation (OPE) based on BPPO. This substitution is vital as it eliminates the need for costly online interactions and enhances safety.
> >
> > Furthermore, in Uni-O4, we replace the value function trained by SARSA with one trained by IQL. This simple replacement, albeit derived from existing methods, plays a crucial role in providing more accurate OPE and advantage function estimations. Additionally, we introduce the ensemble behavior cloning (BC) method with disagreement penalty in order to effectively capture the multi-modality present within the dataset. These seemingly simple components are of paramount importance when it comes to exploring optimal policies.
> >
> > Together, these components within Uni-O4 contribute significantly to the exploration of optimal policies, providing a comprehensive and precise framework for offline RL. This is precisely why Uni-O4 demonstrates significantly superior performance compared to BPPO, even though we replace online evaluation with OPE.
> >
> >
> > **Q3.1: I think there is a slight overclaiming of the experimental results in the Introduction without sufficient comparison of SOTA algorithms. Insufficient comparison of offline RL, including but not limited to [1, 2, 3]:**
> >
> > **A3.1**: Thank you for the suggestion. We have incorporated Diffusion-QL [1], RORL [2], XQL [3], and SQL [4] as additional comparisons in the offline phase. Below are the main findings, with most of the results extracted from the original papers. Results marked with an asterisk (*) indicate that they were reproduced using open-source code and tuned with hyperparameters.
> >
> > Upon inspecting the table below, it becomes evident that Uni-O4 outperforms all other methods in terms of the total score across all tasks. Furthermore, Uni-O4 outperforms all other methods in 16 out of 26 individual tasks. While RORL surpasses Uni-O4 in the total score for MoJoCo Locomotion tasks, it performs worse than Uni-O4 in the other three domains and exhibits limited effectiveness in the Kitchen domain.
> >
> > |Environment|Diffusion-QL[1]|RORL[2]|XQL[3]|SQL[4]|Uni-O4 (Ours)| |
> > |:----|:----|:----|:----|:----|:----|:----|
> > |halfcheetah-medium-v2|51.1|**66.8**|48.3|48.3|52.6 $\pm$ 0.4|
> > |hopper-medium-v2|90.5|**104.8**|74.2|75.5|**104.4 $\pm$ 0.6**|
> > |walker2d-medium-v2|87.0|**102.4**|84.2|84.2|90.2 $\pm$ 1.4|
> > |halfcheetah-medium-replay|47.8|**61.9**|45.2|44.8|44.3 $\pm$ 0.7|
> > |hopper-medium-replay|101.3|**102.8**|100.7|99.7|**103.2 $\pm$ 0.8**|
> > |walker2d-medium-replay|95.5|90.4|82.2|81.2|**98.4 $\pm$ 1.6**|
> > |halfcheetah-medium-expert|96.8|**107.8**|94.2|94.0|93.8 $\pm$ 1.3|
> > |hopper-medium-expert|**111.1**|**112.7**|**111.2**|**111.8**|**111.4 $\pm$ 1.5**|
> > |walker2d-medium-expert|110.1|121.2|112.7|110.0|**118.1 $\pm$ 2.2**|
> > |***locomotion total***|*791.2*|***870.8***|*752.9*|*749.5*|*816.4 $\pm$ 10.5*|
> > |Umaze-v2|93.4|**96.7**|93.8|92.2|93.7 $\pm$ 3.2|
> > |Umaze-diverse-v2|66.2|**90.7**|82.0|74.0|83.5 $\pm$ 11.1|
> > |Medium-play-v2|76.6|76.3|76.0|**80.2**|75.2 $\pm$ 4.4|
> > |Medium-diverse-v2|78.6|69.3|73.6|**79.1**|72.2 $\pm$ 3.8|
> > |Large-play-v2|46.4|16.3|46.5|53.2|**64.9 $\pm$ 2.5**|
> > |Large-diverse-v2|56.6|41.0|49.0|52.3|**58.7 $\pm$ 3.0**|
> > |***Antmaze total***|*417.8*|*390.3*|*420.9*|*431.0*|***448.2 $\pm$ 28.0***|
> > |pen-human|72.8|33.7|85.5|89.2*|**108.2 $\pm$ 10.7**|
> > |hammer-human|4.3* |2.3|8.2|3.8*|**24.7 $\pm$ 4.4**|
> > |door-human|6.9* |3.8|11.5|7.2*|**27.1 $\pm$ 1.3**|
> > |relocate-human|0.0* |0.0|0.2|0.2*|**1.7 $\pm$ 0.6**|
> > |pen-cloned|57.3*|35.7|53.9|69.8*|**101.3 $\pm$ 19.3**|
> > |hammer-cloned|2.1* |1.7|4.3|2.1*|**7.0 $\pm$ 0.9**|
> > |door-cloned|4.1* |-0.1|5.9|4.8*|**10.2 $\pm$ 2.6**|
> > |relocate-cloned|0.0* |0.0|-0.2|-0.1*|**1.4 $\pm$ 0.2**|
> > |***Adroit total***|*147.5* |*77.1*|*169.3*|*177.0*|***281.6 $\pm$ 40.0***|
> > |kitchen-complete|84.0|0.3*|82.4|76.4|**93.6 $\pm$ 2.5**|
> > |kitchen-partial|60.5|0.0*|**73.7**|72.5|58.3 $\pm$ 3.6|
> > |kitchen-mixed|62.6|0.0*|62.5|**67.4**|65.0 $\pm$ 4.6|
> > |***kitchen total***|*207.0*|*0.3*|***218.6***|***216.3***|***216.9 $\pm$ 10.7***|
> > |***Total***|*1563.5* |*1338.5*|*1140.8*|*1573.8*|***1763.1 $\pm$ 89.2***|

---

> > > ### Author Response · Authors · 2023-11-20
> > > **Reviewer Reviewer jjSi (3)**
> > >
> > > **Q3.2: Insufficient comparison (including PROTO, ODT, E2O, SPOT, etc.) or at least discussion of related works on offline-to-online RL. Particularly, the baselines include AWAC, CQL, and IQL, which are naive solutions for offline-to-online RL. PEX presents weak sample efficiency for the above-medium datasets. Cal-ql is not empirically designed for MuJoCo tasks. There is only one relatively strong baseline, i.e., off2on.**
> > >
> > > **A3.2:** Thank you for the valuable suggestion. We consider AWAC, CQL, and IQL as strong baselines for offline-to-online fine-tuning, which are commonly employed in recent works addressing this problem, such as off2on, PEX, Cal-ql, ODT, SPOT, PROTO, E2O, among others. Notably, PEX and Cal-ql represent the latest advancements in offline-to-online fine-tuning methods. Due to time constraints and the extensive number of tasks, it was not feasible to run all methods for comparison. Nonetheless, we made diligent efforts to include as many methods as possible. To ensure a diverse set of baselines, we have included ODT [6] (Decision Transformer family) and SPOT [8] (VAE-based) as additional methods for comparison. Furthermore, we extensively discuss other relevant works [5, 7, 9-12] in the field of offline-to-online RL in the related works section.
> > >
> > > Based on the additional comparisons, the results indicate that Uni-O4 surpasses the state-of-the-art offline and offline-to-online methods in terms of the overall performance across all tasks, as well as in the majority of individual tasks.
> > >
> > > **Q4.1: Minor improvement on MuJoCo tasks in Figure 4. As shown in the figure, off2on significantly outperforms Uni-O4 by a large margin in halfcheetah-medium and halfcheetah-medium-replay. Besides, I also want to point out that a 100 D4RL score already achieves expert-level performance in D4RL benchmark. Thus, further improvement on other settings over 100 is not necessary.**
> > >
> > > **A4.1:** We believe that the relatively small improvement observed in MuJoCo tasks can be attributed to the favorable initialization provided by the offline phase. We demonstrate that the seamless integration of online and offline RL allows for the stable fine-tuning of an initialized policy trained on a sub-optimal dataset, without any decline in performance. This capability showcases the effectiveness of our approach in leveraging offline data to enhance the performance of RL algorithms.
> > >
> > > On the halfcheetah-medium and -medium-replay tasks, off2on outperforms Uni-O4. However, when considering the average performance across all tasks, Uni-O4 outperforms off2on. Notably, off2on exhibits negligible effectiveness in the kitchen and adroit tasks. Overall, Uni-O4 outperforms off2on in 18 out of 21 tasks. On the other hand, Uni-O4 delivers competitive performance on the halfcheetah-medium and -medium-replay tasks when compared with other baseline methods.
> > >
> > > Additionally, We argue that the 100 score defined by D4RL may not necessarily represent the expert-level performance in RL tasks. Many online and offline algorithms are indeed designed with the goal of surpassing the 100 D4RL score. Therefore, we believe that striving for further improvements beyond or approaching the 100 scores is not only reasonable but also necessary to achieve performance levels that truly reflect expert-level capabilities.
> > >
> > > **Q4.2: I also wonder why this work does not consider random datasets, which presents a significant challenge for online finetuning to achieve expert performance.**
> > >
> > > **A4.2:** Thank you for your insightful suggestion. We have added the experiments on the random tasks in **Appendix A.7**. Both the walker2d and hopper tasks can be fine-tuned surpassing 100 D4RL scores. Moreover, the performance of halfcheetah task is competitive compared with other methods.

---

> > > > ### Author Response · Authors · 2023-11-20
> > > > **Reviewer Reviewer jjSi (4)**
> > > >
> > > > **Q5:Comparison in Section 5.2 seems not fair enough. Firstly, I want to know which is the claimed baseline WTW in Figure 5? Additionally, given that IQL is not designed specifically for real-world robotic scenarios, is the comparison between IQL and Uni-O4 fair? (Uni-O4 is revised to adapt to robotic scenarios as stated in the appendix) Maybe a strong baseline can be considered to verify the superiority of Uni-O4.**
> > > >
> > > > **A5:** Thank you for pointing this out. We firmly believe that the comparison remains fair. We consider two baselines in the real-world robot fine-tuning setting. The first baseline is the sim-to-real work called "walk-these-way" [13]. This method focuses on quadruped locomotion and demonstrates the ability to be deployed across various terrains such as grassland, slopes, and stairs, without the need for additional training in each specific environment. This remarkable generalization capability is achieved through extensive randomization of environments and a substantial amount of training data, totaling approximately 2 billion environment steps. However, it should be noted that this method is highly data-inefficient and encounters challenges when attempting to model complex or challenging deployment environments accurately. Thus, we include this baseline in our comparison to highlight the significance of real-world fine-tuning and emphasize the sample efficiency of our online-offline-online paradigm.
> > > >
> > > > The second baseline we consider is IQL, which is an offline-to-online method. We emphasize that IQL is regarded as a strong baseline for real-world robot fine-tining tasks [14]. Several studies [14-17] have utilized IQL for fine-tuning real-world robots in an offline pretraining and online fine-tuning paradigm. In this work, we follow the offline-to-online paradigm of IQL as a baseline, aligning with these previous studies. Additionally, we have provided further implementation details in **Appendix A.3.5** to offer a comprehensive understanding of our approach.
> > > >
> > > > **Q6: I feel a little struggling to follow Section 5.2. Maybe a pseudo-code like A.6 can be provided to make the readers understand the online-offline-online setting more clearly.**
> > > >
> > > > **A6:** Thank you for the valuable suggestion. We have added the pseudo-code of the online-offline-online setting for a comprehensive understanding of Uni-O4 in **Appendix A.3.4**.
> > > >
> > > > **Q7: The experimental results in A.3 make me confusing. I cannot identify obvious differences between Figure 11 (a) and (b).**
> > > >
> > > > **A7:** Thank you for pointing out this. We have highlighted the region of mismatch between the dataset and the policies in Fig. 11. The results show a larger region of mismatch between the single policy and the dataset. Additionally, we aim to provide a clearer visualization to facilitate further investigation into whether ensemble policies contribute to exploring optimal policies. To illustrate this, we have included a motivating example and conducted additional ablation studies in **Appendix A.5**. The results demonstrate that the utilization of multiple diverse behavior policies allows for effective exploration of the high-return action region, particularly in the case of the "medium-replay" dataset, which contains the buffer when the policy is trained with medium performance.
> > > >
> > > > **Q8: Figure 2 is not that intuitive. Maybe more explanations can make it clearer.**
> > > >
> > > > **A8:** Thank you for pointing this out. We have updated Fig. 2 by incorporating sequence numbers and a detailed description for each step. During this stage, policy optimization begins with the learned behavior policies. Specifically, each policy undergoes optimization using the PPO loss for a designated number of gradient steps outlined in step 1 of Fig. 2. Subsequently, both the target policy and behavior policy are assessed using AM-Q to determine whether the behavior policy should be replaced by the target policy, as specified in step 2. If the evaluation results meet the OPE conditions, the behavior policy is substituted with its target policy, denoted in step 3. These three steps are iterated until the predetermined number of gradient steps is reached. Also, we sincerely invite the reviewer to check the GIF of Fig. 2 on our website for a more clear presentation.

---

> > > > > ### Author Response · Authors · 2023-11-20
> > > > > **Reviewer Reviewer jjSi (5)**
> > > > >
> > > > > **Q9:How many seeds and evaluation trajectories for AntMaze tasks in the offline RL setting? Why offline-to-online RL setting does not consider Kitchen, AntMaze, and Adroit-cloned and -relocate tasks?**
> > > > >
> > > > > **A9:** We conducted evaluations of the AntMaze tasks across five different seeds, with each seed consisting of 50 trajectories. Additionally, we have added the experiments on Kitchen and adroit-cloned tasks in **Appendix A.5**. The results show a superior performance compared with the baseline methods. Due to the time constraints and the large number of tasks involved in this study, we did not conduct experiments of offline-to-online RL setting on the AntMaze and -relocate tasks, and we will leave this to future work. It is notable that we have conducted a high volume of tasks on both simulators and real-world robots to evaluate Uni-O4, and it is a common phenomenon in previous studies to not conduct all tasks on D4RL.
> > > > >
> > > > > **Q10:Lack of reproducibility statement.**
> > > > >
> > > > > **A10:** We will release the source code and dataset of real-world robot tasks. Additionally, we have provided detailed information on the hyper-parameters in **Appendix 4.2**.
> > > > >
> > > > > **Q11:Maybe more details on baseline implementation for real-world robot tasks can be provided.**
> > > > >
> > > > > **A11:** Thank you for your valuable suggestion. We have added the detailed implementation of the mentioned two baselines for the real-world robot tasks in **Appendix A 3.4**.
> > > > >
> > > > > **Q12:Why this paper does not provide offline training time for comparison?**
> > > > >
> > > > > **A12:** For our method, the primary runtime of the offline phase is attributed to the supervised learning stage. To evaluate the runtime, we conducted 2 million steps for $Q, V$ training, 0.5 million steps for ensemble policy training, and 1 million steps for transition model training. The comparison of the training time over different methods is non-trivial because it is mainly decided by the training gradient step. For fairness, we conduct the training time comparison by running the open source code with the given training steps.
> > > > >
> > > > > **Q13:Page 4, above Equation 7: dataset $D$ -> $\mathcal{D}$ & Page 9, Hyper-parameter analysis, loss 7 -> Equation 7 is an optimization objective.**
> > > > >
> > > > > **A13:** Thank you for your thorough review. We have made the revisions to the notion and description in the updated version of the manuscript.
> > > > >
> > > > > [1] Wang Z, Hunt J J, Zhou M. Diffusion policies as an expressive policy class for offline reinforcement learning[J]. arXiv preprint arXiv:2208.06193, 2022.
> > > > >
> > > > > [2] Yang R, Bai C, Ma X, et al. Rorl: Robust offline reinforcement learning via conservative smoothing[J]. Advances in Neural Information Processing Systems, 2022, 35: 23851-23866.
> > > > >
> > > > > [3] Garg D, Hejna J, Geist M, et al. Extreme q-learning: Maxent RL without entropy[J]. arXiv preprint arXiv:2301.02328, 2023.
> > > > >
> > > > > [4] Xu H, Jiang L, Li J, et al. Offline rl with no ood actions: In-sample learning via implicit value regularization[J]. arXiv preprint arXiv:2303.15810, 2023.
> > > > >
> > > > > [5] Li J, Hu X, Xu H, et al. PROTO: Iterative Policy Regularized Offline-to-Online Reinforcement Learning[J]. arXiv preprint arXiv:2305.15669, 2023.
> > > > >
> > > > > [6] Zheng Q, Zhang A, Grover A. Online decision transformer[C]//international conference on machine learning. PMLR, 2022: 27042-27059.
> > > > >
> > > > > [7] Zhao K, Ma Y, Liu J, et al. Improving Offline-to-Online Reinforcement Learning with Q-Ensembles[C]//ICML Workshop on New Frontiers in Learning, Control, and Dynamical Systems. 2023.
> > > > >
> > > > > [8] Wu J, Wu H, Qiu Z, et al. Supported policy optimization for offline reinforcement learning[J]. Advances in Neural Information Processing Systems, 2022, 35: 31278-31291.
> > > > >
> > > > > [9] Zheng H, Luo X, Wei P, et al. Adaptive policy learning for offline-to-online reinforcement learning[J]. arXiv preprint arXiv:2303.07693, 2023.

---

> > > > > > ### Author Response · Authors · 2023-11-20
> > > > > > **Reviewer Reviewer jjSi (6)**
> > > > > >
> > > > > > [10] Yu Z, Zhang X. Actor-Critic Alignment for Offline-to-Online Reinforcement Learning[J]. 2023.
> > > > > >
> > > > > > [11] Guo S, Sun Y, Hu J, et al. A Simple Unified Uncertainty-Guided Framework for Offline-to-Online Reinforcement Learning[J]. arXiv preprint arXiv:2306.07541, 2023.
> > > > > >
> > > > > > [12] Ball P J, Smith L, Kostrikov I, et al. Efficient online reinforcement learning with offline data[J]. arXiv preprint arXiv:2302.02948, 2023.
> > > > > >
> > > > > > [13] Margolis G B, Agrawal P. Walk these ways: Tuning robot control for generalization with multiplicity of behavior[C]//Conference on Robot Learning. PMLR, 2023: 22-31.'
> > > > > >
> > > > > > [14] Gürtler N, Blaes S, Kolev P, et al. Benchmarking offline reinforcement learning on real-robot hardware[J]. arXiv preprint arXiv:2307.15690, 2023.
> > > > > >
> > > > > > [15] Zhou G, Ke L, Srinivasa S, et al. Real world offline reinforcement learning with realistic data source[C]//2023 IEEE International Conference on Robotics and Automation (ICRA). IEEE, 2023: 7176-7183.
> > > > > >
> > > > > > [16] Nair A, Zhu B, Narayanan G, et al. Learning on the job: self-rewarding offline-to-online finetuning for industrial insertion of novel connectors from vision[C]//2023 IEEE International Conference on Robotics and Automation (ICRA). IEEE, 2023: 7154-7161.
> > > > > >
> > > > > > [17] Wang J, Dasari S, Srirama M K, et al. Manipulate by Seeing: Creating Manipulation Controllers from Pre-Trained Representations[C]//Proceedings of the IEEE/CVF International Conference on Computer Vision. 2023: 3859-3868.

---

> > > > > > > ### Comment · Reviewer_jjSi · 2023-11-21
> > > > > > >
> > > > > > > Thank you for providing detailed feedback. I acknowledge substantial improvements in this paper, with most of my concerns thoughtfully addressed. Thus, I raise my score to 6.

---

> > > > > > > > ### Author Response · Authors · 2023-11-21
> > > > > > > > **Response to Reviewer jjSi**
> > > > > > > >
> > > > > > > > We are delighted to hear that the reviewer has raised the score! We thank the reviewer again for their invaluable feedback and insights. Meanwhile, we fully agree that the paper is in a much better position as a result of the discussion.

---

### Official Review · Reviewer_bAiM · 2023-10-30

**Soundness:** 3 good
**Presentation:** 3 good
**Contribution:** 2 fair
**Rating:** 6
**Confidence:** 3

**Summary:**

The article introduces Uni-O4, a new method for combining offline and online reinforcement learning. It eliminates redundancy and enhances flexibility by using an on-policy objective for both phases. Uni-O4 employs ensemble policies and a straightforward offline policy evaluation approach in the offline phase to address mismatches between behavior policy and data. The approach leads to better offline initialization and efficient online fine-tuning for real-world robot tasks and achieves state-of-the-art results in various simulated benchmarks.

**Strengths:**

1) Despite some minor flaws, this paper is written in a standardized and organized manner, allowing people to quickly capture the core innovative points and ideas of the paper.

2) The Uni-O4 framework proposed in the article unifies the learning objectives of online and offline learning, making the transition from offline learning to online learning smoother.

3) This method has shown excellent performance in various experiments and has also achieved good results in real-world machine experiments.

**Weaknesses:**

1）The behavior cloning method proposed in section 3.1 requires training multiple policy networks, which incurs significant computational overhead. At the same time, it does not mention how to get $\hat{pi}_{\beta}$ from a policy set.

2）Definition error, the definition of f used in formulas 6 and 7 is incorrect. Taking the maximum value of multiple distributions cannot guarantee a single distribution (the sum cannot be guaranteed to be 1), and analysis based on this definition is also meaningless. If the code is truly implemented based on this definition, I am skeptical about the final performance of the algorithm.

3) The proposed offline strategy evaluation method relies on the accuracy of the probability transfer model T, and using the transfer model for evaluation will introduce more errors.

4) The entire method has made too many approximations to the problem and lacks corresponding error analysis.

5） The legend in Figure 3 is missing to know the correspondence between curves and algorithms.

**Questions:**

1) Can you provide a detailed reconstruction method for policy $\hat{\pi}_{\beta}$, whether to select any one from the policy set $\Pi_n$ or integrate it using the f function to obtain a policy?

2) Is there a way to evaluate the quality of behavior cloning? Can you compare your proposed method of behavior cloning with previous methods?

3) Can we analyze the errors in the approximate part? You can cite the results of previous work to prove it. For this article, you do not need to prove the size of the approximation error. You only need to quantify the approximation error to a certain extent, analyze the potential impact, and find ways to avoid negative effects.

---

> ### Author Response · Authors · 2023-11-20
> **Response to Reviewer bAiM (1)**
>
> Thanks Reviewer bAiM for the valuable review of our paper. We appreciate the questions you raised and are committed to delivering a comprehensive response to address the issues.
>
> **Q1: The behavior cloning method proposed in section 3.1 requires training multiple policy networks, which incurs significant computational overhead.**
>
> **A1:** Thank you for your comment. We have thoroughly examined the computational overhead and performance implications associated with different ensemble sizes of policies in Section 5.3 and **Appendix A.11**. Our findings indicate that an ensemble size of 4 strikes a balance between performance and computational overhead. Specifically, the total training time for the offline phase is approximately 265 minutes (with an ensemble size of 4), compared to 200 minutes (with a single policy) using our PyTorch implementation. Despite the slight increase in training time, this trade-off is acceptable considering the significant performance improvement achieved through the ensemble approach.
>
> **Q2: Definition error, the definition of f used in formulas 6 and 7 is incorrect. Taking the maximum value of multiple distributions cannot guarantee a single distribution (the sum cannot be guaranteed to be 1), and analysis based on this definition is also meaningless. If the code is truly implemented based on this definition, I am skeptical about the final performance of the algorithm.**
>
> **A2:** Thank you for pointing this out. We have taken measures to address this concern. Firstly, we have normalized the distribution of the combined policies, as outlined in Proposition 1. Additionally, we have derived a lower bound over the defined distance, as described in Theorem 1. By optimizing this lower bound, we are able to enhance the diversity among behavior policies. Notably, the derived lower bound corresponds to the penalty term mentioned in Equation 7 of the previous version. Therefore, the implementation can be guaranteed based on these derived results.

---

> > ### Author Response · Authors · 2023-11-20
> > **Response to Reviewer bAiM (2)**
> >
> > **Q3: The entire method has made too many approximations to the problem and lacks corresponding error analysis. & The proposed offline strategy evaluation method relies on the accuracy of the probability transfer model T, and using the transfer model for evaluation will introduce more errors.**
> >
> > **A3:** Thank you for this suggestion. We have reevaluated the approximation employed throughout the entire pipeline and included a thorough analysis to address this concern. Firstly, we have revised the notion of value functions trained by IQL manner form $Q_{\pi_{\beta}}$ and $V_{\pi_{\beta}}$ to $\widehat{Q_{\tau}}$ and $\widehat{V_{\tau}}$. It is helpful to distinct the value functions trained by SARSA and IQL which offers an optimal solution $Q^*$ under the dataset constraints [1], i.e., $\lim_{\tau \rightarrow 1} Q_{\tau} (s, a) = Q^{\ast} (s, a)$. $\widehat{Q_{\tau}}$ and $\widehat{V_{\tau}}$ are used to express the value functions obtained through gradient-based optimization. This revision helps to make more clear error analysis.
> >
> > During the multi-step policy optimization stage, the decision for behavior policy replacement is based on the defined metric AM-Q, which incorporates the true transition model $T$ and optimal value function $Q^*$. In the offline setting, however, the agent does not have access to the true model and $Q^*$. But $Q_{\tau}$ approaches $Q^*$ based on the dataset support constraints. We fit $\hat{T}$ and $\widehat{Q_{\tau}}$ by gradient-based optimization. In this way, the practical AM-Q can be expressed as $\widehat{J_{\tau}} (\pi)$. Thus, the OPE bias is mainly coming from the transition model approximation. We have analyzed and derived the bound of the offline policy evaluation, described by Theorem 2 in Section 3.2, please refer to check out the details.
> >
> > BPPO has derived the offline monotonical improvement bound (their Theorem 3) between the target policy and behavior policy, i.e., $\pi_{k+1}$ and $\pi_k$. However, since the behavior policy is updated iteratively, this bound can result in cumulative errors that disrupt the desired monotonicity. Consequently, this bound cannot be directly utilized in the offline setting. This limitation is the reason why BPPO requires online evaluation to ensure performance improvement when replacing the behavior policy with the target policy. Given the OPE bound, we can replace the online evaluation with AM-Q to guarantee monotonicity.
> >
> > Additionally, we compute the advantage function by $(\widehat{Q_{\tau}}-\widehat{V_{\tau}})$, which provides an approximation of the optimal $Q^*$ and $V^*$ based on the dataset constraint assumption. In **Appendix A.8**, we conduct experiments to demonstrate this replacement is a superior choice compared to iteratively fitting the value function of the target policy in the offline setting. The results from these experiments prove that iteratively updating the value function will lead to overestimation which causes unstable training performance and potential crashes.

---

> > > ### Author Response · Authors · 2023-11-20
> > > **Response to Reviewer bAiM (3)**
> > >
> > > **Q4: The legend in Figure 3 is missing to know the correspondence between curves and algorithms.**
> > >
> > > **A4:** Thank you for pointing this out. We have addressed this issue in the revised paper.
> > >
> > > **Q5: Can you provide a detailed reconstruction method for policy $\hat{\pi}_{\beta}$, whether to select anyone from the policy set $\prod_n$ or integrate it using the f function to obtain a policy?**
> > >
> > > **A5:** In this work, we leverage the learned diverse behavior policies to capture the multi-modality in the dataset. During the offline multi-step policy optimization stage, each behavior policy is optimized by the PPO objective and guided by the AM-Q OPE metric. Thus, the behavior policy is not selected from the ensemble or using a function to obtain it. On the other hand, sampling the action from mixed policies or other policies conflicts with the PPO objective, as it aims to restrict the distribution distance between the target policy and behavior policy. In other words, the target policy needs to be close to the behavior policy because PPO is an on-policy algorithm that incorporates important sampling techniques.
> > >
> > > **Q6: Is there a way to evaluate the quality of behavior cloning? Can you compare your proposed method of behavior cloning with previous methods?**
> > >
> > > **A6:** Thank you for sharing your insightful suggestion. In this work, our motivation for utilizing a diverse ensemble of policies is to effectively capture the multi-modality inherent in the dataset, thereby enabling exploration of the high-return action region. Evaluating the quality of the ensemble behavior cloning directly is challenging, as our primary objective is to improve the final performance of the offline multi-step policy optimization process. Therefore, in Section 5.3, we evaluate the offline performance of the proposed ensemble behavior cloning approach with a disagreement penalty, comparing it to the standard behavior cloning (BC) method and the ensemble BC approach without the regularization term. The hyper-parameter analysis demonstrates that the proposed ensemble BC with disagreement penalty, when properly tuned, outperforms the alternative methods. Additionally, we have conducted an ablation study to answer the "Do diverse policies help in exploring the high-return action region to discover optimal policies" in **Appendix A.5**. The results show that it is helpful for exploring the optimal policies.

---

> ### Author Response · Authors · 2023-11-22
> **Further Discussion**
>
> Dear reviewer bAiM:
>
> We appreciate again for your constructive comments and helpful suggestions. Since the Reviewer-Author discussion phase is coming to an end, we would like to post a follow-up discussion.
>
> In our previous response, we clarified the raised questions and made corresponding improvements in the updated manuscript. We hope to further discuss with you whether your concerns have been addressed. We are always looking forward to your further comments or suggestions.

---

### Official Review · Reviewer_rY3g · 2023-11-01

**Soundness:** 3 good
**Presentation:** 3 good
**Contribution:** 3 good
**Rating:** 8
**Confidence:** 4

**Summary:**

This paper proposes a new algorithm called Uni-O4 that unifies offline and online reinforcement learning using an on-policy optimization approach. The key ideas are:
- Using an on-policy PPO objective for both offline and online learning to align the objectives.
- In the offline phase, using an ensemble of policies and offline policy evaluation to safely achieve multi-step policy improvement.
- Seamlessly transferring between offline pretraining and online fine-tuning without extra regularization or constraints.
- Evaluating Uni-O4 on both simulated tasks like Mujoco and real-world quadruped robots.

**Strengths:**

- Simple and unified design without needing extra regularization or constraints for stability. Avoids issues like conservatism or instability in prior offline-to-online methods.
- Impressive results surpassing SOTA on offline RL and offline-to-online tasks. Significantly boosts offline performance and enables rapid, stable online fine-tuning.
- Policy ensemble provides good coverage over offline data distribution. Offline policy evaluation enables safe multi-step improvement.
- Excellent results on real-world robots - pretraining, offline adaptation, online finetuning. Showcases efficiency and versatility.

**Weaknesses:**

- The complexity of the method, especially regarding the ensemble behavior cloning and disagreement-based regularization, may present a steep learning curve for practitioners.

**Questions:**

- What are the computational overheads associated with the ensemble policies, and how do they impact the method's scalability?
- Why don't use the ensemble approach to mitigate mismatches instead of other methods for handling the diverse behaviors in the datasets? For example, Diffusion-QL [1] demonstrates that Diffusion model can be used to learn multimodal policy.

[1] Wang, Zhendong, Jonathan J. Hunt, and Mingyuan Zhou. "Diffusion Policies as an Expressive Policy Class for Offline Reinforcement Learning." In The Eleventh International Conference on Learning Representations. 2022.

---

> ### Author Response · Authors · 2023-11-20
> **Response to Reviewer rY3g**
>
> Thank you, Reviewer rY3g, for your valuable review of our paper. We sincerely appreciate the questions you have raised and we are fully dedicated to providing a comprehensive response to address all of the concerns.
>
> **Q1: The complexity of the method, especially regarding ensemble behavior cloning and disagreement-based regularization, may present a steep learning curve for practitioners.**
>
> **A1:** We have added a more intuitive explanation and algorithm description, aiming to provide a clearer understanding for practitioners. Uni-O4 adopts a straightforward approach by combining online and offline RL learning without introducing additional regularization during their transfer.
>
> In detail, the online RL algorithm employed is a standard PPO. While the offline stage involves multiple components such as value functions, transition models, and ensemble BC, it is worth noting that these components are designed to be implementation-friendly, straightforward, and efficient. For the ensemble BC, we train each policy using a standard loss, incorporating an approximate KL-divergence penalty with a combined policy. This penalty term is implemented by sampling data from the offline dataset. These components remain fixed during the multi-step policy optimization stage.
> During this stage, AM-Q consists of the well-trained transition model, and the $Q$-function is used to evaluate the performance of the target policy when replacing the behavior policy. By permitting the replacement of the current behavior policy with the target policy, it achieves multi-step optimization. It differs from iteratively updating the behavior policy which leads to accumulated errors that disrupt the desired monotonicity. In contrast, the multi-step method can guarantee monotonicity due to the AM-Q error is bounded.
>
> We'll also fully open-source our code with detailed documentation to help implement our method, including in a new real-world robot scenario.
>
>
> **Q2: What are the computational overheads associated with the ensemble policies, and how do they impact the method's scalability?**
>
> **A2:** Thank you for this suggestion. We've conducted the computational overheads and the performance associated with the ensemble size of policies in Section 5.3 and **Appendix A.11**. We found that the ensemble size $4$ is a trade-off between performance and the computational overheads. Specifically, the whole training time of the offline phase is around 265 minutes (ensemble size $4$) vs. 200 minutes (single policy) based on Pytorch implementation. It's acceptable due to the significant performance improvement. For offline training, the primary runtime of the offline phase is attributed to the supervised learning stage. To evaluate the runtime, we conducted 2 million steps for $Q, V$ training, 0.5 million steps for ensemble policy training, and 1 million steps for transition model training. However, in practical scenarios, these training steps can be halved, resulting in reduced runtime.
>
> **Q3: Why don't use the ensemble approach to mitigate mismatches instead of other methods for handling the diverse behaviors in the datasets? For example, Diffusion-QL [1] demonstrates that the Diffusion model can be used to learn multimodal policy.**
>
> **A3:** Thank you for the insightful comment. Diffusion-QL [1] is a representative work that uses diffusion policies to handle the multi-modal dataset. It uses a diffusion model as the policy to predict actions and trains the policy with a behavior cloning loss and an extra state-action value term. It is a value-based training paradigm. Indeed, the diffusion policy has the capability to replace ensemble BC during the supervised stage. However, it cannot directly be optimized by the PPO objective during the multi-step policy optimization stage. This kind of policy-based algorithm needs to model the policy's distribution explicitly and rely on differentiating through the policy distribution to compute gradients and perform updates. It is not suitable for optimizing this kind of generative model directly.
>
> Additionally, we have added diffusion-QL as a baseline for comparison. The performance of Uni-O4 significantly outperforms the Diffusion-QL over several domain tasks. Moreover, Uni-O4 is more computationally efficient than Diffusion-QL, consuming 12.6 hours (Diffusion-QL) vs. 4.4 hours (Uni-O4).
> | Environment/method | Diffusion-QL [1]    | Uni-O4    |
> | ------------------ | ------- | ------- |
> | MuJoCo locomotion average           | 88.0    | 90.7   |
> | AntMaze average   | 69.6    | 74.7    |
> | Adroit-Pen average    | 65.1    | 108.1    |
> | kitchen average  | 69.0    | 72.3    |
> | All average | *72.9* | *86.4* |
>
> [1] Wang Z, Hunt J J, Zhou M. Diffusion policies as an expressive policy class for offline reinforcement learning[J]. arXiv preprint arXiv:2208.06193, 2022.

---

> ### Author Response · Authors · 2023-11-22
> **Further Discussion**
>
> Dear reviewer rY3g:
>
> We would like to thank you again for your constructive comments and helpful suggestions. Since we are nearly at the end of the discussion phase, we'd like to know whether our response has addressed your concerns and we are always looking forward to your further comments or suggestions.

---

### Official Review · Reviewer_CZhd · 2023-11-05

**Soundness:** 3 good
**Presentation:** 3 good
**Contribution:** 3 good
**Rating:** 8
**Confidence:** 3

**Summary:**

The authors propose a new approach, Uni-O4, to combine offline and online reinforcement learning, which is an important and challenging problem in the field. Uni-O4 can effectively address the mismatch issues between the estimated behavior policy and the offline dataset,  and it can achieve better offline initialization than other methods and be more stable for the later online fine-tuning phase. The experimental results on several benchmark tasks show that Uni-O4 outperforms existing state-of-the-art methods in terms of stability, final performance, and the capability for real-world transferring.

**Strengths:**

1. Uni-O4 can seamlessly transfer between offline and online learning, enhancing the flexibility of the learning paradigm.

2. The experiments are sufficient and persuasive. The experiments on real-world robots showed very good performance in the provided videos.

**Weaknesses:**

1. Notions are confusing in this paper, especially after the overloading in Equ. (8).

2. In Fig.2, It is hard to capture the Offline Multi-Step Optimization process, i.e. the sequence relationship of each step.

3. In Sec 3.1:  "BPPO leads to a mismatch ... due to the presence of diverse behavior policies in the dataset D",  could authors explain further why the diversity is blamed for the mismatch?

4. Lack of theoretical analysis (to support the motivation of technique details), but it has sufficient experiments thus this point is acceptable I think.

**Questions:**

Suggest to add legends for Fig. 3 or bringing the legend in Fig. 4 forward.

---

> ### Author Response · Authors · 2023-11-20
> **Response to Reviewer CZhd (1)**
>
> Thanks Reviewer CZhd for the valuable review of our paper. We appreciate the questions you raised and are committed to delivering a comprehensive response to address the issues.
>
> **Q1: Notions are confusing in this paper, especially after the overloading in Equ. (8).**
>
> **A1:** Thank you for pointing this out. We have rewritten Section 3.2 "Multi-step policy ensemble optimization" to enhance clarity in our presentation. In Equ. 8 (7 in the new version), specifically, we overload the notion of behavior policy for introducing the iteration number. It is necessary for showcasing how Uni-O4 performs multi-step optimization by adding the iteration number as subscript $k$ in behavior policies $\pi^i_k$ where $i$ specifies the policy index in the ensemble. Furthermore, we have revised the notion of the $Q$ and $V$ -function to be consistent with IQL as $\widehat{Q_{\tau}}$ and $\widehat{V_{\tau}}$ from $Q_{\pi_{\beta}}$ and $V_{\pi_{\beta}}$. Because we found that the previous version ($Q_{\pi_{\beta}}$ and $V_{\pi_{\beta}}$) will cause ambiguity. On the other hand, IQL has the capability to reconstruct the optimal value function, i.e., $\lim_{\tau \rightarrow 1} Q_{\tau} (s, a) = Q^{\ast} (s, a)$  based on dataset support constraints [1]. In this work, we exploit the desirable property to facilitate multi-step policy optimization and recover the optimal policy. $Q_{\tau}$ and $V_{\tau}$ are the optimal solution. We use $\widehat{Q_{\tau}}$ and $\widehat{V_{\tau}}$ to denote the value functions obtained through gradient-based optimization.
>
> **Q2: In Fig.2, It is hard to capture the Offline Multi-Step Optimization process, i.e. the sequence relationship of each step.**
>
> **A2:** Based on this comment, we have updated Fig. 2 by incorporating sequence numbers and a detailed description for each step. During the offline multi-step optimization stage, the policy optimization begins with the learned behavior policies. Specifically, each policy undergoes optimization using the PPO loss for a designated number of gradient steps outlined in step 1 of Fig. 2. Subsequently, both the target policy and behavior policy are assessed using AM-Q to determine whether the behavior policy should be replaced by the target policy, as specified in step 2. If the evaluation results meet the OPE conditions, the behavior policy is substituted with its target policy, denoted as step 3. These three steps are iterated until the predetermined number of gradient steps is reached. Also, we sincerely invite the reviewer to check the GIF of Fig. 2 on our website for a clearer presentation.

---

> ### Author Response · Authors · 2023-11-20
> **Response to Reviewer CZhd (2)**
>
> **Q3: In Sec 3.1: "BPPO leads to a mismatch ... due to the presence of diverse behavior policies in the dataset D", could authors explain further why the diversity is blamed for the mismatch?**
>
> **A3:** One example that can illustrate this motivation is the presence of multi-modality within a diverse dataset. In other words, the dataset exhibits multiple modes, with the main mode primarily composed of low-return actions. Conversely, the subdominant mode consists of low-density but high-return actions. In such a scenario, standard behavior cloning (BC) is susceptible to imitating the high-density but low-return actions, resulting in a bias towards fitting the main mode. However, during the offline multi-step optimization stage, the policy optimization is constrained by the clip function, making it difficult for the policy to escape this mode. Consequently, this can lead to a sub-optimal policy as it becomes unable to explore the high-return action region.
>
> In contrast, our ensemble BC approach learns diverse behavior policies that are more likely to cover all modes present in the dataset. This facilitates exploration of the high-return region, enabling the discovery of the optimal policy. To support this claim, we have included a motivating example and conducted additional ablation studies in **Appendix A.5**. These findings demonstrate that the utilization of multiple diverse behavior policies allows for effective exploration of the high-return action region, particularly in the case of the "medium-replay" dataset, which contains more sub-optimal data than others.
>
> **Q4: Lack of theoretical analysis (to support the motivation of technique details), but it has sufficient experiments thus this point is acceptable I think.**
>
> **A4:** Thank you for this suggestion. We have reevaluated the approximation employed throughout the entire pipeline and included a thorough analysis to address this concern. During the multi-step policy optimization stage, the decision for behavior policy replacement is based on the proposed AM-Q metric, which incorporates the true transition model $T$ and optimal value function $Q^*$. In the offline setting, however, the agent does not have access to the true model and $Q^*$. But $Q_{\tau}$ approaches $Q^*$ based on the dataset support constraints. We fit $\hat{T}$ and $\widehat{Q_{\tau}}$ by gradient-based optimization. In this way, the practical AM-Q can be expressed as $\widehat{J_{\tau}}(\pi)$. Thus, the OPE bias is mainly coming from the transition model approximation. We have analyzed and derived the bound of the offline policy evaluation, described by Theorem 2 in Section 3.2, please refer to check out the detail.
>
> BPPO [2] have derived the offline monotonical improvement bound (their Theorem 3) between the target policy and behavior policy, i.e., $\pi_{k+1}$ and $\pi_k$. However, since the behavior policy is updated iteratively, this bound can result in cumulative errors that disrupt the desired monotonicity. Consequently, this bound cannot be directly utilized in the offline setting. This limitation is the reason why BPPO requires online evaluation to ensure performance improvement when replacing the behavior policy with the target policy. Given the OPE bound, we can replace the online evaluation with AM-Q to guarantee monotonicity.
>
> Additionally, we compute the advantage function by $(\widehat{Q_{\tau}}-\widehat{V_{\tau}})$, which provides an approximation of the optimal $Q^*$ and $V^*$ based on the dataset constraint assumption. In **Appendix A.8**, we conduct experiments to demonstrate this replacement is a superior choice compared to iteratively fitting the value function of the target policy in the offline setting. The results from these experiments prove that iteratively updating the value function will lead to overestimation which causes unstable training performance and potential crashes.
>
> **Q5: Suggest to add legends for Fig. 3 or bringing the legend in Fig. 4 forward.**
>
> **A5:** Thank you for pointing this out. We have addressed this issue in the updated manuscript.
>
> [1] Kostrikov, I., Nair, A., & Levine, S. (2021). Offline reinforcement learning with implicit q-learning. arXiv preprint arXiv:2110.06169.
>
> [2] Zhuang Z, Lei K, Liu J, et al. Behavior proximal policy optimization[J]. arXiv preprint arXiv:2302.11312, 2023.

---

> > ### Comment · Reviewer_CZhd · 2023-11-21
> > **Thanks for authors' explanation**
> >
> > Thank you for this careful feedback, which addressed most of my concerns. After reading all responses to me and other reviewers,  I am delighted to raise my score from 6 to 8.

---

> > > ### Author Response · Authors · 2023-11-21
> > > **Response to Reviewer CZhd**
> > >
> > > We are very happy to hear that the reviewer has raised the score! We thank the reviewer again for their invaluable feedback and insights.

---

### Author Response · Authors · 2023-11-20
**Response to all reviewers**

We appreciate all the reviewers for their thoughtful feedback and constructive suggestions.  We would like to thank **Reviewer CZhd** for acknowledging that 'Uni-O4 can seamlessly transfer between offline and online learning, enhancing the flexibility of the learning paradigm.', **Reviewer rY3g** thinks that uni-O4 is a "Simple and unified design without needing extra regularization" with "excellent results on real-world robots - pretraining, offline adaptation, online finetuning" and "showcases efficiency and versatility", **Reviewer bAiM** think our paper is well-written, "allowing people to quickly capture the core innovative points and ideas of the paper", and **Reviewer jjSi** evaluates this work as "a decent paper."

In response to the valuable feedback from the reviewers, we have diligently carried out extensive additional experiments and included in-depth discussions to incorporate their insightful comments into our work. To provide a concise summary for the reviewers, we present below the major updates that have been incorporated in the revised submission:
1. **Analysis of the AM-Q evaluation error:** In section 3.2, we have derived the bound of the proposed AM-Q OPE metric, referring to Theorem 2. Given this and the offline monotonical improvement bound from BPPO [1], Uni-O4 can replace the online evaluation with AM-Q to guarantee monotonicity.

2. **Extra ablation study on ensemble policies:** In Appendix A.5, we first give a simple motivating example when the dataset exhibits multiple modes, the ensemble policies are more likely to cover all modes present in the dataset. This facilitates exploration of the high-return region, enabling the discovery of the optimal policy. Also, we have conducted experiments to investigate whether the learned diverse policies can explore the high-return region.

3. **Extra baseline comparison and more tasks are added:** We have added more strong baselines during offline [2-5] and offline-to-online phases [6, 7] to evaluate the superiority of Uni-O4. Moreover, we have added the Kitchen and adroit-cloned tasks for offline-to-online fine-tuning settings in Appendix A.7.

4. **The comparison between $\widehat{Q_{\tau}}$ and $Q_{\pi_k}$ in Uni-O4:** In Appendix A.8, we conduct experiments to demonstrate that the usage of $\widehat{Q_{\tau}}$ which approximate the optimal $Q^*$ under the dataset support constraints, is a superior choice compared to iteratively fitting the value function of the target policy in the offline setting. The results from these experiments prove that iteratively updating the value function will lead to overestimation which causes unstable training performance and potential crashes.

5. **Training time comparison and other revision:** In Appendix A.11, we have conducted computational overhead ablation on ensemble size. Also, we compare the training time of offline and online fine-tuning phases. We revised Fig.1 and Fig.2 for a clearer presentation.

6. **Open-source code claim:** We'll fully open-source our code with detailed documentation to help implement our method, including real-world setups.

[1] Zhuang Z, Lei K, Liu J, et al. Behavior proximal policy optimization[J]. arXiv preprint arXiv:2302.11312, 2023.

[2] Wang Z, Hunt J J, Zhou M. Diffusion policies as an expressive policy class for offline reinforcement learning[J]. arXiv preprint arXiv:2208.06193, 2022.

[3] Yang R, Bai C, Ma X, et al. Rorl: Robust offline reinforcement learning via conservative smoothing[J]. Advances in Neural Information Processing Systems, 2022, 35: 23851-23866.

[4] Garg D, Hejna J, Geist M, et al. Extreme q-learning: Maxent RL without entropy[J]. arXiv preprint arXiv:2301.02328, 2023.

[5] Xu H, Jiang L, Li J, et al. Offline rl with no ood actions: In-sample learning via implicit value regularization[J]. arXiv preprint arXiv:2303.15810, 2023.

[6] Zheng Q, Zhang A, Grover A. Online decision transformer[C]//international conference on machine learning. PMLR, 2022: 27042-27059.

[7] Wu J, Wu H, Qiu Z, et al. Supported policy optimization for offline reinforcement learning[J]. Advances in Neural Information Processing Systems, 2022, 35: 31278-31291.

---

### Meta-Review · Area_Chair_y3Xp · 2023-12-11

**Metareview:**

In this paper, a new algorithm which aims for unifying offline and online reinforcement learning is proposed.  The proposed algorithm learns multiple policies in the offline setting for multi-step policy improvement, and then the policy will be used for online fine-tuning.

Although there are several concerns raised by the reviewers, most of the reviews are positive.

**Justification For Why Not Higher Score:**

There are several issues actually I think the authors did not addressed well:

1, The proposed method is not well-motivated. I do not see why the authors insist to avoid regularization.

2, The ensemble policies learning is both memory and computation cost.

In fact, these drawbacks make the algorithm only applicable for small models. In practical applications inn both robotics and LLM, regularized behavior cloning + on-policy improvement will dominate the proposed algorithm due to the memory complexity.

3, Moreover, the exploration issue, which is the key problem in online RL, has never been discussed.

**Justification For Why Not Lower Score:**

All reviewers suggest to accept the paper.

---

### Decision · Program_Chairs · 2024-01-16

Accept (poster)